# Interplay between coding and non-coding regulation drives the Arabidopsis seed-to-seedling transition

Benjamin J. M. Tremblay [1], Cristina P. Santini[1], Yajiao Cheng[1], Xue Zhang[2], Stefanie Rosa [2] & Julia I. Qüesta [1] ✉

Translation of seed stored mRNAs is essential to trigger germination. However, when RNAPII re-engages RNA synthesis during the seed-to-seedling transition has remained in question. Combining csRNA-seq, ATAC-seq and smFISH in *Arabidopsis thaliana* we demonstrate that active transcription initiation is detectable during the entire germination process. Features of non-coding regulation such as dynamic changes in chromatin accessible regions, antisense transcription, as well as bidirectional non-coding promoters are widespread throughout the Arabidopsis genome. We show that sensitivity to exogenous ABSCISIC ACID (ABA) during germination depends on proximal promoter accessibility at ABA-responsive genes. Moreover, we provide genetic valida-tion of the existence of divergent transcription in plants. Our results reveal that active enhancer elements are transcribed producing non-coding enhancer RNAs (eRNAs) as widely documented in metazoans. In sum, this study defining the extent and role of coding and non-coding transcription during key stages of germination expands our understanding of transcriptional mechanisms underlying plant developmental transitions.

The transition from dormant dry seeds to a vegetative seedling is an irreversible process that requires a near-complete reprogramming of the plant transcriptome. In *Arabidopsis thaliana* (Arabidopsis), seed dormancy relies primarily on the phytohormone abscisic acid (ABA). Sensitivity to ABA is maintained via the activity of the protein DELAY OF GERMINATION 1 (DOG1), which permits phosphorylation and activation of transcription factors involved in ABA signaling such as ABA INSENSITIVE 5 (ABI5)[1–3]. Dormancy release can occur via moist chilling (stratification), which triggers a gradual increase in the pro-duction of the signaling hormone gibberellic acid (GA) alongside a reactivation of cellular metabolism as a result of rehydration[4,5]. During the initial stages of germination, growth is driven by cellular expansion as a result of GA signaling[6]. Crucially, in this period the plant remains susceptible to growth arrest by exogenous ABA (leading to an upre-gulation of ABI5) until the transition to post-germinative growth and

cellular division[7]. However, the regulatory aspects of the loss of ABA sensitivity remains unclear.

Seed maturation at the cellular level is marked by chromatin compaction and reduction in nuclear size, as well as dehydration and a general conversion of metabolism towards accumulation seed reserves[8,9]. Stored mRNAs in dry seeds are an absolute requirement for germination, with cyclohexamide treatments to inhibit translation being sufficient to completely prevent it[10,11]. However, the ability of dry seeds to undergo transcription (as opposed to simply storing mRNAs) has remained in question. Although compaction is generally asso-ciated with transcriptional repression[12], previous experiments have hinted that transcription factories may be present in dry seeds[10,13–15]. While several studies have characterized the changes in the Arabi-dopsis transcriptome during germination, these could not distinguish between nascently produced and seed-stored mRNAs[2,4,16–20]. Detailed

[1]Centre for Research in Agricultural Genomics (CRAG), CSIC-IRTA-UAB-UB, Campus UAB, Bellaterra, Barcelona, Spain. [2]Plant Biology Department, Swedish University of Agricultural Sciences (SLU), Uppsala, Sweden. ✉e-mail: julia.questa@cragenomica.es

experimental data of nascent transcriptome changes over the course of germination remain absent as of yet.

The strong condensation of the Arabidopsis genome taking place during seed maturation is reversed over the course of germination[9]. This suggests significant epigenetic changes affecting transcription are required during germination. Indeed, global changes in DNA methylation during germination contribute to the regulation of many genes[16,18,21]. Profiling chromatin accessibility during germination using Assay for Transposase-Accessible Chromatin using sequencing (ATAC-seq) has revealed a large number of differentially accessible chromatin regions (ACRs) associated with changing gene expression[22]. In addition, several studies have noted the role of long non-coding RNAs (lncRNAs) in regulating transcription of protein coding genes during germination[23–25]. Understanding the role of lncRNAs during the seed-to-seedling transition remains limited by their under-representation in typical transcriptome profiling methods such as RNA-seq. One technique used to overcome this is sequencing nascent RNAs.

In order to clarify the role of nascent transcription and its regulation by non-coding RNAs during the seed-to-seedling transition, we performed a time course of capped-small RNA sequencing (csRNA-seq)[26] and ATAC-seq[27]. In combination these methods provided an unprecedented view of the dynamics of transcription initiation and promoter accessibility over the course of the seed-to-seedling transition, revealing transcription events associated with key cis-elements. By applying single molecule imaging, we demonstrated that active RNA synthesis takes place at all stages of germination, from early imbibition onward. The combined csRNA-seq and ATAC-seq approach was suitable to explore the regulation of important biological processes such as promoter sensitivity to ABA. A large number of previously unknown non-coding RNA initiation events were observed and associated with affecting nearby coding RNA initiation. We identified and characterized Arabidopsis bidirectional promoters, providing experimental validation of divergent transcription as well as transcription arising from enhancer elements. Our work clarifies the prevalence of previously unknown features of the regulation of transcription in plants.

## Results

### Active transcription is detectable during all stages of Arabidopsis germination

To investigate regulation of transcription initiation during the seed-to-seedling transition, we performed csRNA-seq, sRNA-seq and total RNA-seq of Arabidopsis Col-0 samples. We used RNA extracted from dry seeds (DS), stratified seeds (after 24 h (S24) and 72 h (S72)), and after 6 h (L6), 26 h (L26; 50% radicle emergence) and 57 h (L57; fully expanded green cotyledons) of exposure to light (Fig. 1a). Most non-coding RNAs are rapidly degraded by the exosome complex. In order to allow us to reconstruct additional putative lncRNAs associated with novel transcription start site (TSS) clusters from our csRNA-seq, we additionally sampled mutants in genes related to this pathway including *HUA ENHANCER2* (*hen2-4*) and *RIBOSOMAL RNA PROCESSING 4* (*rrp4-2*) at the fully expanded green cotyledon stage, confirming the capture of unstable non-coding RNAs by our csRNA-seq (see Supplementary Note 1)[28,29]. We also profiled chromatin accessibility of Col-0 at a subset of these time-points using ATAC-seq (Fig. 1a). All replicates for all samples and methods had high Pearson correlation values (ranging from 0.9 to 0.99) and the correlation between time-points showed a clear progression (Supplementary Fig. 1a–c). Similar trends were observed using principal component analysis (PCA; Fig. 1b). The stratified samples were more distant from the DS samples along PC1 in the csRNA-seq as compared to the PC1 of the RNA-seq, suggesting significant changes in transcription initiation as a result of imbibition. To test this, we performed single molecule RNA FISH (smFISH) using probes targeting both spliced and unspliced RNA of a gene with apparent transcriptional activity across the entire seed-to-seedling

transition (Supplementary Fig. 1e). Strikingly, we observed clear transcription sites within the nuclei of cells in all stages of germination (Fig. 1c), confirming that the csRNA-seq is capturing true transcriptional activity and that transcription occurs even during the earliest stages of imbibition. While we could also detect sites of transcription initiation in the DS csRNA-seq data these likely represent partially degraded or initiated RNAPII transcripts from seed maturation, though our data cannot disprove that some rare events of active transcription may be occurring at some TSSs (see Supplementary Note 2).

Across all of our csRNA-seq samples (including *hen2-4* and *rrp4-2*) we detected a total of 30,273 TSS clusters (of which 21,470 were assigned to nearby existing Araport11 annotations), alongside 30,213 ACRs across all of our ATAC-seq samples (Supplementary Data 1; Supplementary Data 2)[30]. The median number of significantly differentially expressed (DE) TSSs between consecutive time-points was 8643, with 16,514 DE TSSs (55% of all detected TSSs) between the DS and L57 samples (Supplementary Data 3). We compared read density across the 19,260 protein coding genes with a detected TSS. Heatmaps demonstrate a clear strong enrichment of reads at the TSS of genes in the csRNA-seq as compared to the sRNA-seq and RNA-seq, coinciding with the region of increased accessibility in the ATAC-seq sample (Fig. 1d; Supplementary Fig. 1f–i; see Supplementary Note 1 for additional discussion). Using different expression thresholds, we detected an increasing number of active TSSs over the course of germination which could not be captured by the RNA-seq alone (Supplementary Fig. 1j–m). The remaining detected non-coding TSSs could be distinguished by their differing promoter base composition, with the exception of core promoter elements such as the TATA box and initiator (Inr) element (Supplementary Fig. 1n–q). In addition, the signal enrichment in the csRNA-seq across all types of TSSs had an improved TSS-to-gene signal ratio when compared to GRO-cap, matching those of CAGE data from Arabidopsis seedlings (Supplementary Fig. 1r)[31,32]. Thus, the sensitivity of the csRNA-seq method provides an improved view of the extent of transcription during seed germination.

### Gene regulatory programs during the seed-to-seedling transition

The csRNA-seq and ATAC-seq datasets capture stage-specific regulatory programs (Fig. 2a, b) with 89% of the detected TSSs organized into 6 clusters, each cluster generally corresponding to each of the 6 time-points (Fig. 2a). 62% of the detected ACRs were organized into 5 clusters (Fig. 2b). Three of these were time-point specific: A1 corresponded to DS, A2 to L6, and A5 to L57. The other two were shared between time-points: A3 with L6 and L26, and A4 with L26 and L57. To compare the overlap between the two datasets, we assigned each ACR to its nearest detected TSS and observed significant overlap between temporally matching clusters (Fig. 2c). The DS-specific clusters exhibited unique properties: C1 had the largest fraction of protein-coding TSSs (as well as the lowest overall percentage of non-coding transcription), and A1 had the largest fraction of ACRs overlapping transposable elements (Fig. 2d, e; Supplementary Fig. 2a). These data suggest additional yet-unknown layers of transcriptional regulation may exist in dry seeds. Clustering analysis of our matching total RNA-seq dataset showed a similar set of sample-specific clusters as found in the csRNA-seq (Supplementary Fig. 2b–d).

To determine which regulatory programs were being detected by the csRNA-seq, gene ontology (GO) enrichment of the cluster genes was performed (Fig. 2a; Supplementary Data 4). Genes related to cellular catabolism, vesicle-mediated transport, seed development and response to ABA were enriched in the DS-specific cluster C1. The earliest stratification sample (S24, C2) contained enriched terms for genes related to RNA splicing. The final stratification time-point (S72, C3) was most enriched for terms associated with a response to a decrease in oxygen levels and RNA metabolic process, underscoring the apparent

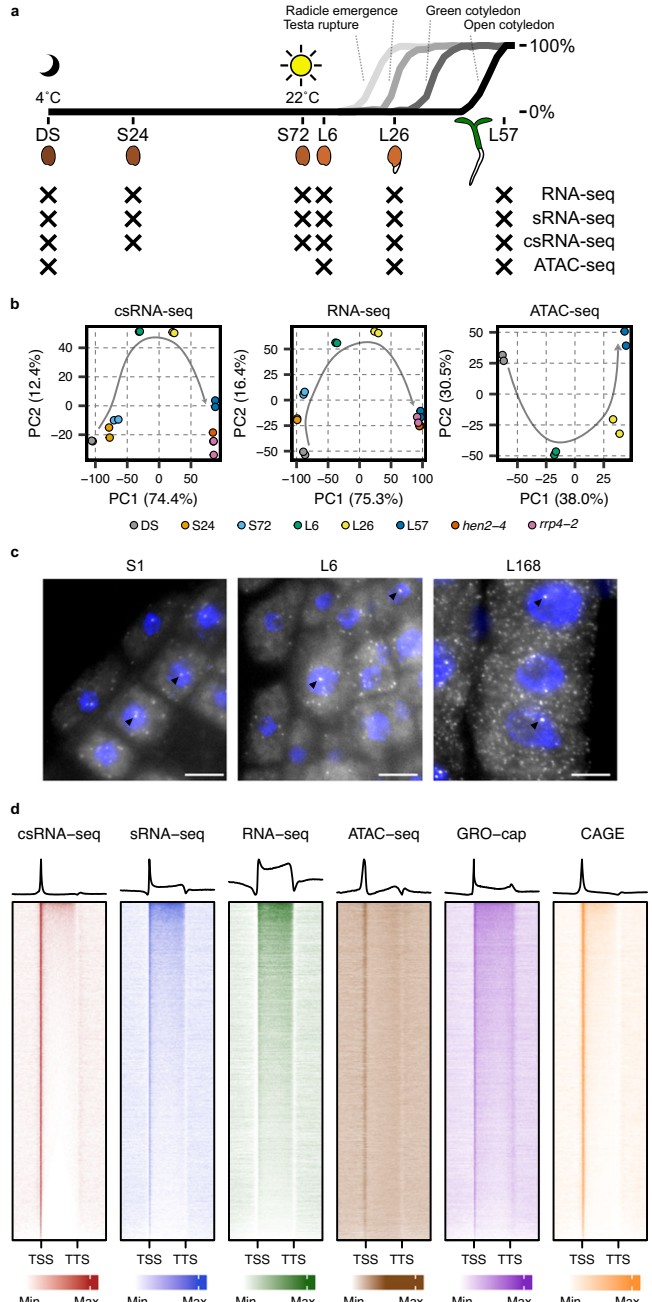

**Fig. 1 | Profiling transcription initiation during the seed-to-seedling transition.**
**a** Schematic overview of the sampled time-points for the csRNA-seq, sRNA-seq, RNA-seq and ATAC-seq experiments overlaid on data showing the timing of key markers of the seed-to-seedling transition in Col-0 seed. In total 6 time-points were selected: dry seeds (DS), seeds stratified in water for 24 h (S24) and 72 h (S72) in the dark at 4 °C, and germinating seeds 6 h (L6), 26 h (L26), and 57 h (L57) after being moved to the light at 22 °C. **b** Principal component analyses of all samples for the csRNA-seq, RNA-seq and ATAC-seq datasets. Two additional genotypes are included which were sampled at the same time as L57: *hen2-4* and *rrp4-2*. **c** smFISH in 1 h imbibed seeds (S1), germinating seeds 6 h after being moved to the light (L6), and 7 d seedling roots (L168) using probes for the unspliced RNA of a gene showing csRNA-seq expression during all time-points (*AT1G04170*; Supplementary Fig. 1e). The scale bar represents 10 μm. Brighter spots in the nucleus (see arrows) represent active transcription sites in germinating seeds, whereas smaller bright dots in the surrounding area likely indicate spliced transcripts bound by a smaller number of exonic-only probes (28 / 48 total probes). An RNase control image of the S1 sample is shown in Supplementary Fig. 1d. Experiments were repeated independently at least two times. **d** Heatmaps and average metagene plots showing the presence of read density across all detected protein coding genes in the L57 time-point of the csRNA-seq, sRNA-seq, RNA-seq and ATAC-seq. These data are compared to previously published GRO-cap and CAGE data obtained from Arabidopsis seedlings[31,32]. Read density is scaled independently for each dataset between the 0th and 90th percentiles (shown as min and max, respectively).

These results demonstrate that, although seed-stored mRNAs are important, all stages of germination are accompanied by active transcription of their component mRNAs, including those of the translation machinery required to trigger germination (L6).

## ABA sensitivity during germination is controlled by DNA accessibility

Having confirmed that our dataset captured known transcriptional events during the seed-to-seedling transition, we next sought to uncover the role of cis-element accessibility and transcription initiation. We performed motif discovery analyses on two sets of sequences: (1) the promoter sequences around each detected TSS (−400 bp, +100 bp), and (2) 500 bp around the center of each ACR. We successfully discovered a core set of motifs enriched in both sets of sequences (Fig. 3a), as well as additional motifs only enriched in one of the two (Supplementary Fig. 3a, b). The core set of common motifs resembled known cis-elements, including the ABA-responsive element (ABRE) or G-box element[34], drought-responsive element (DRE) or low-temperature responsive element (LTRE)[35], ACGT element (ACE)[36], Telo-box[37], and Site II motif[38]. We assigned possible transcription factors acting upon these elements based on their expression levels and cluster membership (Fig. 3a; Supplementary Data 7; see Methods). Two of the most highly enriched cis-elements (M1 and M4) were matching with ABI5 and RELATED TO AP2 1 (RAP2.1), both of which are expressed and active within the DS. Also highly enriched were the Telo-box (M23) and Site II motif (M2) in L6-associated clusters (mostly in translation-associated genes), however none of the TFs expressed in the C4 cluster had similar binding sites (Supplementary Data 7). Despite this, these two motifs likely play important roles in the activation of translation-related genes during germination as the Site II motif and Telo box are strongly enriched within the center of the promoter ACR and at the start of transcription initiation, respectively (Supplementary Fig. 3c–e). These data demonstrate the strong link between chromatin accessibility, TF binding and transcription initiation during the seed-to-seedling transition.

As a way to test whether our combined expression and chromatin accessibility data could be used to answer meaningful biological questions, we next sought to understand the transcriptional basis for the early germination window in which growth arrest can be induced using exogenous ABA. This process has previously been linked to ABI5 activity[7]. Examining the expression levels and promoter accessibility of

importance of RNA metabolism and alternative splicing during germination[17,18]. Matching the L6 time-point, cluster C4 was nearly singularly strongly enriched for translation-related terms. Interestingly, this cluster also contained the fewest transcription factors (TFs) of all the clusters (Supplementary Fig. 2e–f). The transition to post-germinative growth (L26, C5), was enriched for genes related to DNA replication and protein glycosylation. An increase in transcription related to DNA replication matches the beginning of cell division associated with post-germinative growth. Furthermore, protein glycosylation has been associated with the decline in ABA sensitivity during germination[33]. Finally, the L57-specific cluster C6 was enriched for a multitude of different seedling and growth pathways, including photosynthesis, cell wall biogenesis, carbohydrate biosynthetic process, and root morphogenesis. This cluster also contained the largest number of TFs of all the clusters. The GO enrichment analysis was also performed with the ATAC-seq and total RNA-seq clusters and largely revealed similar enriched terms (Fig. 2b; Supplementary Data 5; 6).

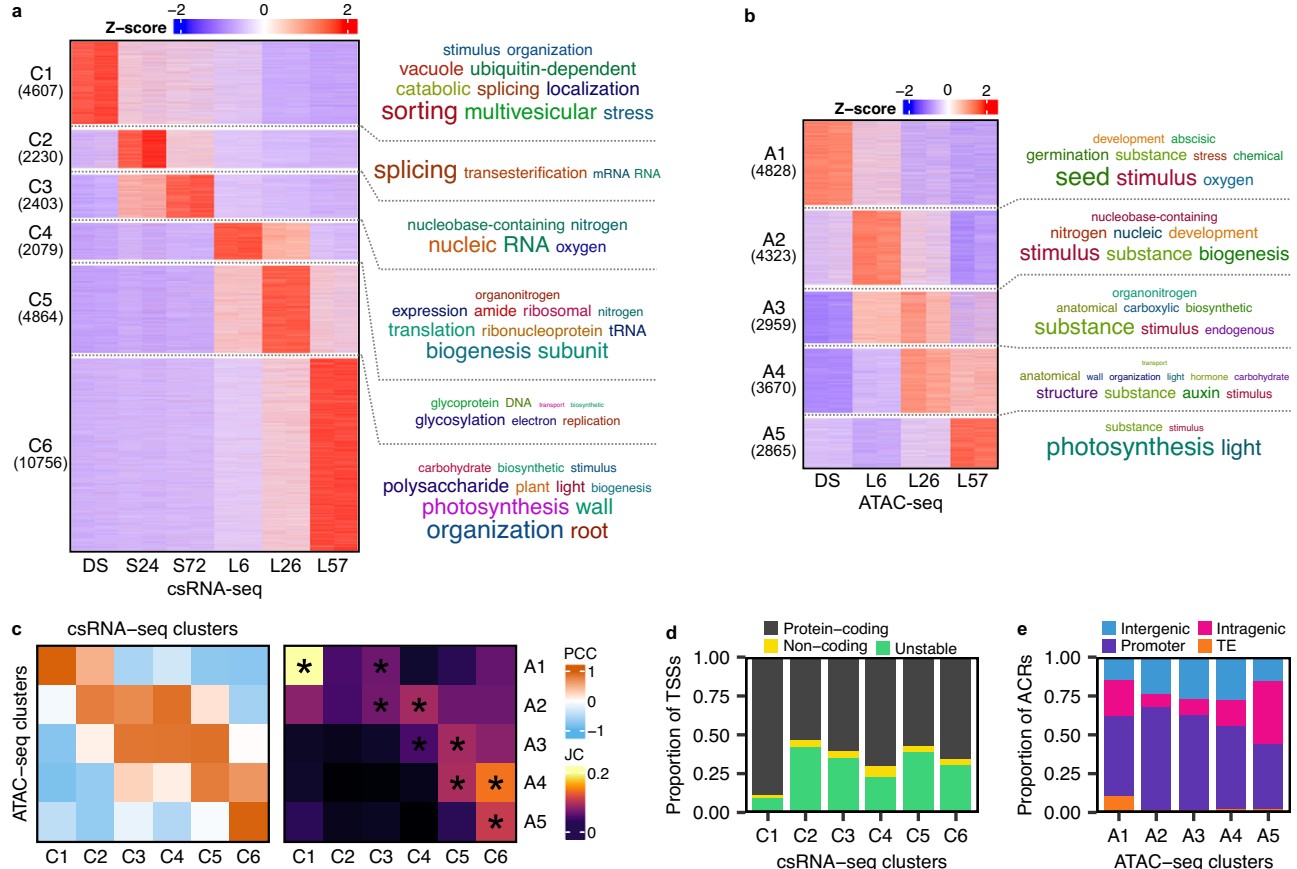

**Fig. 2 | Combined csRNA-seq and ATAC-seq analysis reveals dynamics of gene regulatory programs and non-coding transcription during the seed-to-seedling transition. a** Heatmap of developmental clusters from the csRNA-seq time-series. Rows represent Z-scores of the expression of individually annotated TSSs. Associated genes were enriched for overrepresented gene ontology terms, followed by an individual keyword enrichment analysis to generate word clouds of overrepresented keywords to the right of the heatmap, with their size being proportional to the level of enrichment. **b** Heatmap of developmental clusters from the ATAC-seq time-series. Rows represent Z-scores of the accessibility of individual ACRs. A word cloud of enriched keywords from a gene ontology enrichment analysis for ACR-associated genes is shown on the right. **c** Comparison analyses of the csRNA-seq and ATAC-seq clusters. The left heatmap shows the Pearson correlation coefficient between the average Z-score profiles of each cluster. The right heatmap shows the Jaccard coefficient of the number of common associated genes of each cluster. Comparisons with significant overlap are marked with an asterisk (P-value < 10⁻⁶). Significance testing was performed using Fisher's exact test without correction for multiple testing. **d** Proportion of annotated TSS types in the csRNA-seq clusters. **e** Proportion of annotated ACR types in the ATAC-seq clusters.

possible ABI5 targets, we observed that while the average expression level of these genes undergoes a sharp drop from the DS to the S24 time-points, their promoter accessibility loss occurs much more slowly (Fig. 3b). This is in contrast to the L6-specific genes regulated by the Site II motif: both their expression and promoter accessibility drop quickly upon reaching post-germinative growth (Fig. 3c). To illustrate this, we examined more closely the expression and promoter accessibility of *ABI5* and two well characterized ABI5 targets: *LATE EMBRYOGENESIS ABUNDANT 1* (*EM1*)[39] and *6* (*EM6*)[40]. While *EM1* and *EM6* undergo a sharp (greater-than 10-fold) decrease in expression upon stratification, their accessibility remains elevated by the time of germination (L6) and only decays after this point (Fig. 3d). Similarly *ABI5* expression levels are nearly completely abolished at the time of germination, yet its promoter accessibility remains high and only shows a slow decay. These observations match the window of time in which the germinating seedling remains sensitive to ABA. One possibility is that exogenous ABA treatment may sufficiently reactivate residual ABI5 protein to bind still-accessible targets, until the point at which they have lost all accessibility during post-germinative growth and thus no longer allow ABI5 binding. Using a previously published MNase-seq dataset from leaves confirmed this, as a nucleosome can be observed in the promoters of DS-specific genes and more generally at instances of the motif itself, likely blocking access for TFs and RNAPII to initiate

transcription (Supplementary Fig. 3f–g)[41]. This supports the link between the gradual loss of DNA accessibility and the reduction in ABA sensitivity during the post-germinative transition.

## Evidence of substantial non-coding transcription during germination

Our work thus far demonstrates that RNAPII engages in RNA synthesis as early as 1 h post imbibition (Fig. 1d) with our csRNA-seq data showing highly dynamic reprogramming of protein coding genes at all stages of germination (Figs. 1b, 2a; Supplementary Fig. 2d). Furthermore, by evaluating the dynamics of proximal promoter accessibility at ABA-responsive genes we provide a model for the timing of seedling response to ABA during germination (Fig. 3b, d; Supplementary Fig. 3f) which could not have been predicted by only looking at the levels of transcripts produced (by either csRNA-seq or RNA-seq). We next wished to examine changes in the non-coding transcriptome from our data. Making use of the csRNA-seq method's ability to capture non-coding transcription initiation irrespective of transcript stability (percent non-coding transcription ranging from 7 to 17%, compared to approximately 10% in GRO-cap data and less than 1% in CAGE; Supplementary Fig. 4a)[31,32], we first sought to critically evaluate the discovered non-coding TSSs. In total 10,106 non-coding TSSs were discovered, of which 1293 could be assigned to existing Araport11

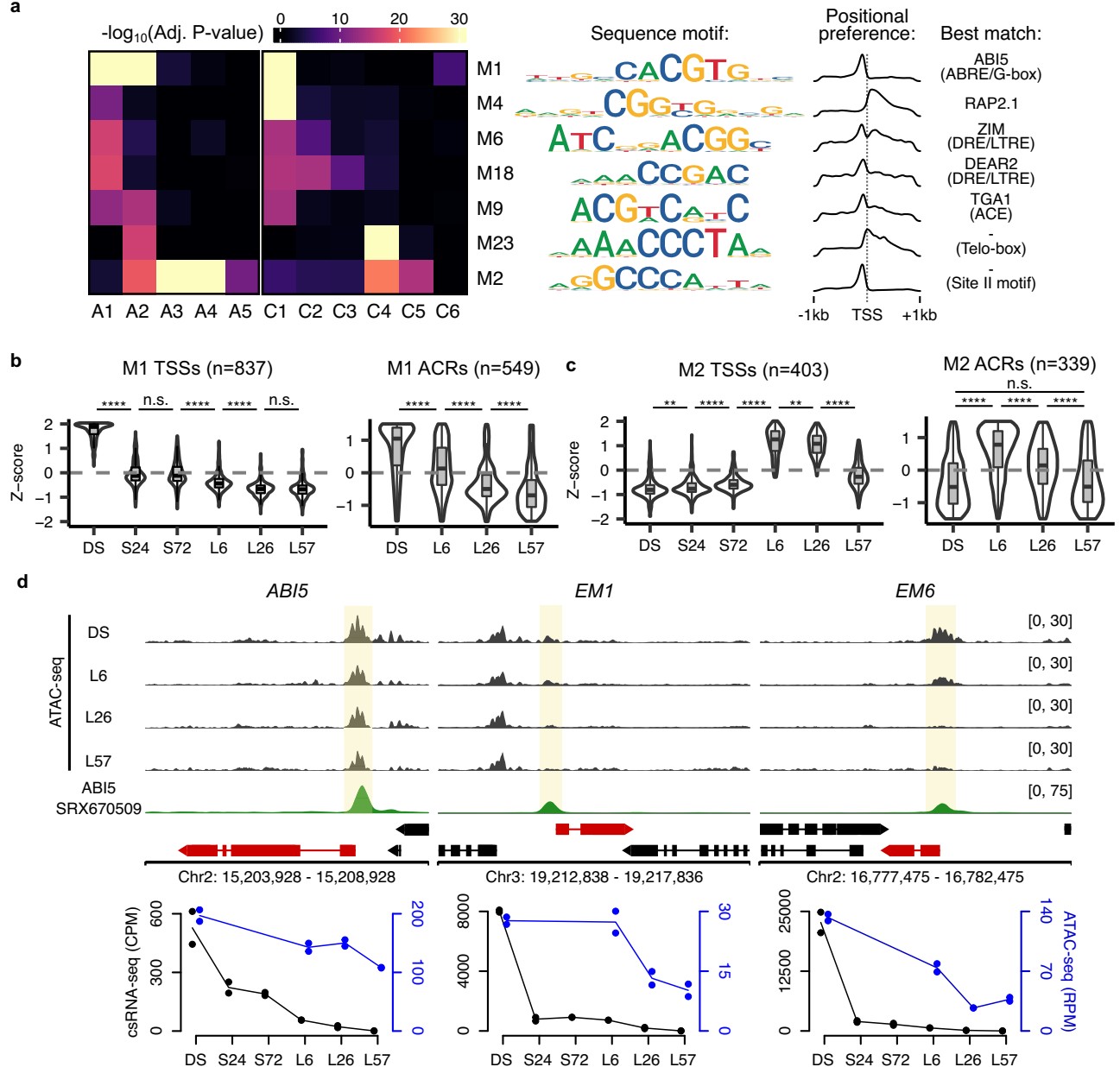

**Fig. 3 | The timing of ABA sensitivity during germination is regulated by promoter DNA accessibility. a** Enrichment of discovered motifs from the promoters of TSSs found in the csRNA-seq clusters as well as the ACRs found in the ATAC-seq clusters. A heatmap shows the level of enrichment ($-\log_{10}$ adjusted $P$-value) of the motif in each cluster, with each row representing a unique motif (shown to the right using an information content motif logo). The density of the motifs is shown to demonstrate their positional preference in promoters. The best matching known binding transcription and/or element name is included on the right. $P$-values were calculated using one-sided Fisher's exact tests with FDR correction for multiple testing. **b, c** Violin plots of Z-scores of csRNA-seq and ATAC-seq data for TSSs and

their overlapping ACRs, respectively, containing the M1 or M2 motifs. The lower, middle and upper hinges correspond to first quartile, median, and third quartile, respectively. The lower and upper whiskers extend to the minimal/maximal value respectively or 1.5 times the interquartile range, whichever is closer to the median. $P$-values were calculated using two-sided Mann–Whitney tests with Holm correction for multiple testing (n.s. = not significant, **$p < 0.01$, ****$p < 0.0001$). **d** ATAC-seq and ABI5 DAP-seq[113,114] read coverage density tracks for the DS, L6, L26 and L57 time-points for the genes *ABI5*, *EM1*, and *EM6* (units in RPM). Below these are plotted the corresponding csRNA-seq (in black) and ATAC-seq (in blue) quantification data for the respective TSSs and promoter-associated ACRs.

annotations (Fig. 4a)[30]. Of the unknown 8813 TSSs, we successfully reconstructed 2588 putative lncRNAs, with 1689 intergenic lncRNAs and 899 antisense lncRNAs (Fig. 4b). An additional 2702 antisense non-coding TSSs and 2841 intergenic non-coding TSSs were found for which no predicted lncRNA transcript could be reconstructed (herein named unstable TSSs). We next tested whether these TSSs exhibited properties similar to known Araport11 non-coding RNAs. The fraction of TSSs with an upstream TATA box and recognizable Inr element was very similar, alongside clear evidence of conservation in other plants

using PhyloP and PhastCons scores (Fig. 4c; Supplementary Fig. 4b–d)[42]. TSSs without a reconstructed transcript had lower conservation levels immediately downstream of the TSS compared to those with a putative lncRNA, suggesting a link between sequence conservation and transcript stability. The reconstructed lncRNAs also had similar size distributions, expression levels in the csRNA-seq and RNA-seq, coding potential and expression variability across samples (Supplementary Fig. 4e–j)[43]. We also detected similar patterns of enrichment of RNAPII and the active histone marks H3K4me3 and

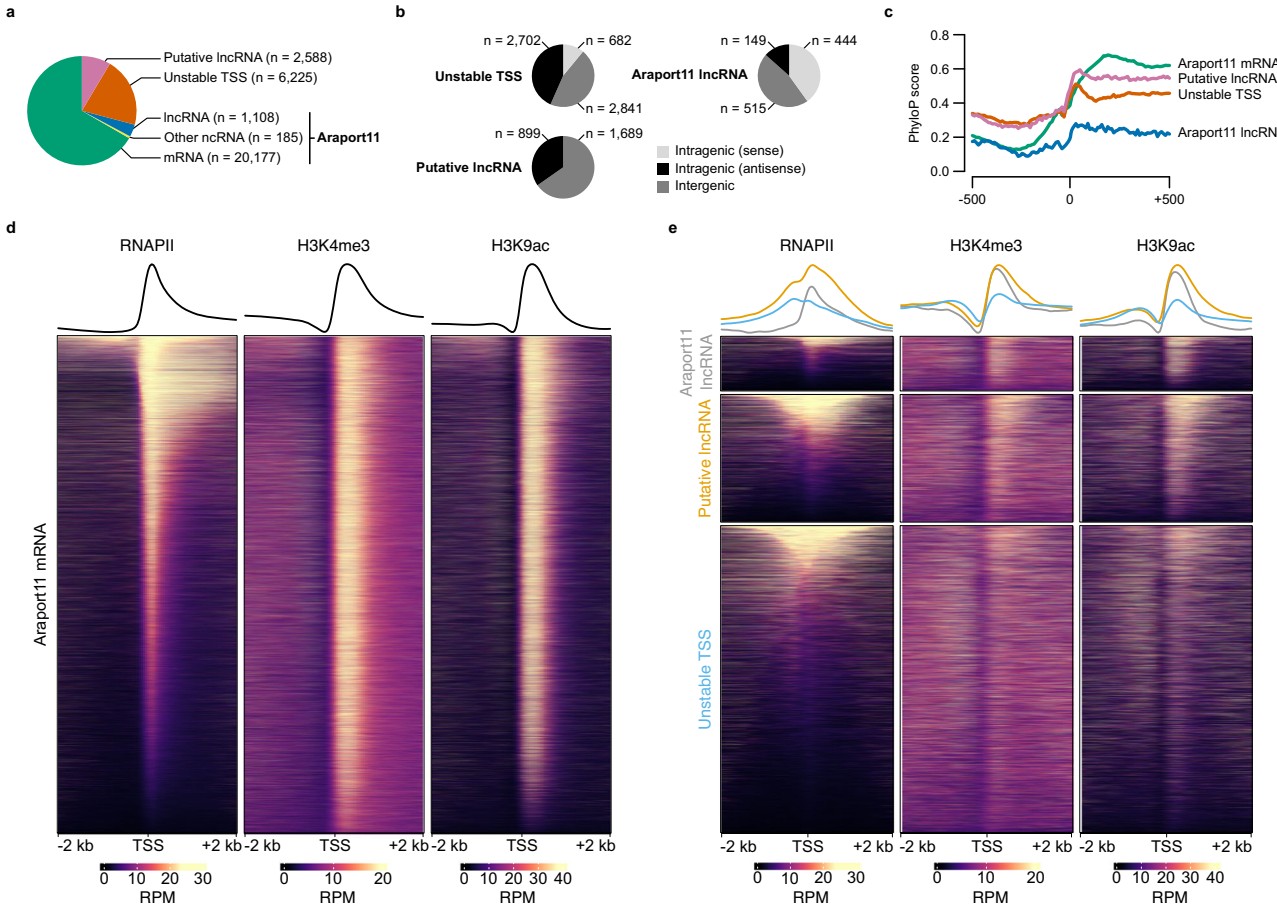

**Fig. 4 | Characteristics of non-coding transcription during the seed-to-seedling transition. a** Pie chart of annotated TSS types from the csRNA-seq. TSSs associated with an existing Araport11 TSS were annotated as either mRNA, lncRNA or Other ncRNA. The remaining TSSs were annotated as either Putative lncRNA (when a putative transcript could be reconstructed from the RNA-seq) or Unstable TSS (when transcript reconstruction was not possible). **b** Pie charts of the positional contexts of non-coding TSSs (lncRNA, Putative lncRNA and Unstable TSS). Features which did not overlap any protein coding gene were annotated as Intergenic. Features which overlapped a protein coding gene were annotated as Intragenic,

and as either sense or antisense in brackets to denote the relative orientation to the overlapping gene. **c** Average conservation of promoters by annotated TSS type (mRNA, lncRNA, Putative lncRNA, and Unstable TSS), using PhyloP scores calculated from 63 plant species[42]. The coverage of scores is from 500 bp upstream and downstream of the primary TSS coordinate. **d** Heatmaps, and accompanying average profiles of external RNAPII[110], H3K4me3[115], and H3K9ac[116] ChIP-seq datasets from Arabidopsis seedlings of the 4 kb area around protein coding TSSs. **e** A repeat of the plotting from (**g**), instead showing the non-coding annotated TSS types.

H3K9ac at both of our protein coding and non-coding TSSs (Fig. 4d, e). Together, the data above confirmed that csRNA-seq is an excellent method to explore dynamics of non-coding transcription. In the following sections, we exploit this technique to explore features of non-coding regulation during germination.

**Regulation of gene expression by antisense transcription**
To investigate the effect of non-coding TSSs on nearby protein coding genes, we looked at the possible contribution of antisense transcription (an important regulatory mechanism in Arabidopsis). Of the 19,260 protein-coding genes with at least one detected TSS, 20.7% contained an antisense non-coding TSS in our csRNA-seq data (Fig. 5a; Supplementary Data 8). We also successfully observed previously described antisense lncRNAs, such as *COOLAIR* (antisense to the *FLOWERING LOCUS C* (*FLC*) gene) and *asDOG1* (including a new TSS for the latter; Supplementary Fig. 5a, b)[23,44]. While most of our observed antisense TSSs were close to the transcription termination site (TTS) of genes, we observed a significant minority remained within close proximity of the gene TSS, irrespective of gene length, at a median distance of 492 bp (Fig. 5b; Supplementary Fig. 5c). We labeled the former case as "distal" antisense TSSs, and the latter as "proximal" (Fig. 5c). Rates of transcription initiation were generally

lower at antisense TSSs when compared to the sense TSS, with a median ratio of 8:1 sense:antisense expression (Supplementary Fig. 5d). Surprisingly, we found that antisense TSSs showed levels of sequence conservation nearly comparable to that of the sense TSS, though this was much less pronounced for proximal antisense TSSs (Supplementary Fig. 5e, f). The sequence composition of distal antisense TSSs also resembled that of the sense TSS much more than for proximal antisense TSSs (Supplementary Fig. 5g–j). Distal antisense TSSs also had comparably more accessible promoters (Supplementary Fig. 6a). Accumulation of the active histone marks H3K4me3 and H3K9ac were also present at antisense TSSs, though to a lesser degree than observed at sense TSSs (Supplementary Fig. 6b, c).

To understand the role that these antisense TSSs may be playing in regulating the expression of sense TSSs, we compared the expression patterns of genes with antisenses. These genes had slightly higher coefficients of variation across germination, suggesting fewer constitutive genes have antisense transcription (Supplementary Fig. 6d). Furthermore, the presence of a proximal antisense TSS generally correlated with lower levels of expression of the sense TSS as well as reduced levels of RNAPII accumulation over the gene body when compared to the high levels of RNAPII detected in genes with distal

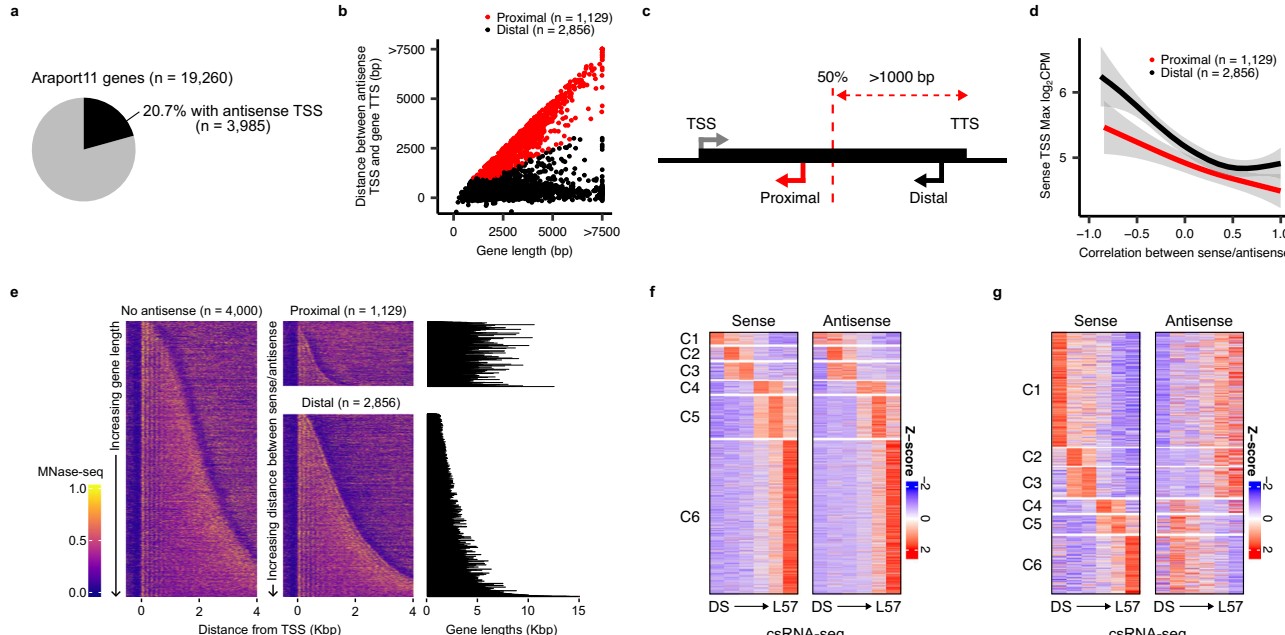

**Fig. 5 | Global antisense transcription regulates gene expression during germination. a** Pie chart of the fraction of active protein coding genes with detected antisense csRNA-seq signal. **b** Plot of the distance of antisense TSSs from the primary gene TSS against the length of the gene. Each dot represents an individual gene-antisense pair, colored based on the following antisense classification: antisense TSSs within the first half of the gene body, as well as at least 1 kb from the gene TTS, are labeled as proximal (red). All others are marked as distal (black). **c** Schematic overview of the classification method for distinguishing proximal and distal antisense transcription, using the classification system from (**b**). **d** Smoothed average max $\log_2$ expression level of sense TSSs with an associated proximal or distal antisense plotted against the Pearson correlation coefficient between the expression of each sense/antisense pair. Smoothing was performed using a

generalized additive model. The shaded area represents the 95% confidence interval of the model. **e** Heatmaps of external MNase-seq data from leaves at a random sample of 4000 active protein coding genes without a detected TSS (sorted by gene size), as well as genes with a proximal or distal antisense (sorted by sense/antisense inter-TSS distance). The gene lengths for genes with a detected antisense are plotted as horizontal bar plots to the right. MNase-seq[41] read density is row-normalized between 0 and 1. **f** Heatmap of Z-scores of csRNA-seq quantification for sense and antisense TSSs for genes with a detected antisense TSS and a positive correlation between their expression (Pearson correlation coefficient greater than 0.5), split by the csRNA-seq cluster membership of the sense TSS. **g** Same as for (**f**), instead plotting genes where the correlation between the sense/antisense pairs are less than −0.25.

antisense TSSs (Fig. 5d; Supplementary Fig. 6e–g). One possible explanation for this difference is that the presence of an active proximal TSS may disrupt the positioning of nucleosomes within the gene body; indeed, these genes generally had fewer, less distinct nucleosomes, likely preventing normal traversal of RNAPII across the gene (Fig. 5e; Supplementary Fig. 6h). In addition, we observed that genes in which there was a positive correlation between the expression patterns of sense and antisense TSSs had lower average maximum expression (Fig. 5d). A closer examination of when correlated and anticorrelated cases were expressed during germination revealed that the former were overrepresented in the final L57 stage (for example, the *PIN-FORMED 4* (*PIN4*) antisense), whereas the latter were overrepresented in the DS stage (for example, the *RESPONSE TO DESICCATION 26* (*RD26*) antisense; Fig. 5f, g; Supplementary Fig. 6i–k). Since the DS stage likely has the smallest number of distinct transcriptionally active cell types (which do not undergo cell division until L26), and vice versa for the L57 stage, these data suggest that sense and antisense transcription likely cannot occur simultaneously and only appear to correlate in our bulk sequencing data as a result of their expression in different tissues. Therefore, our results favor the hypothesis that sense and antisense transcription are mostly mutually exclusive as shown before for the antisense *COOLAIR*[45].

## Divergent transcription is not coordinated in Arabidopsis

Due to the high sensitivity of detection of unstable non-coding transcription events of the csRNA-seq, we next sought to answer whether we could observe bidirectional transcription, a question that remains contested in the study of Arabidopsis transcription. Although some studies have found no evidence of this[31,46], others have more recently

claimed otherwise[32,47–49]. From our combined csRNA-seq data, in total we found 1127 protein coding (pcTSS)-pcTSS, 1643 non-coding TSS (ncTSS)-pcTSS, and 381 ncTSS-ncTSS opposite-facing pairs within 500 bp of each other (Fig. 6a; Supplementary Data 9). In total we detected 48% and 61% of the ncTSS-pcTSS pairs observed by ref. 47 and ref. 32, respectively, as well as 12% and 6% of the ncTSS-ncTSS pairs observed by ref. 32 and ref. 48 (Supplementary Fig. 7a, b). There was no clear relationship between pcTSS-pcTSS distance and correlation, but we observed an increase in average correlation with decreasing distance for ncTSS-pcTSS and ncTSS-ncTSS pairs, leading us to conclude that pcTSSs likely cannot function from a shared promoter simultaneously while ncTSSs (in combination with another pcTSS or ncTSS) may have the potential to do so (Fig. 6b).

We first focused on the evidence for divergent transcription (ncTSS-pcTSS pairs). We could observe a clear sharing of a single ACR by ncTSS-pcTSS pairs within 500 bp (with a median distance of 207 bp), which was lost at greater inter-TSS distances (Fig. 6d; Supplementary Fig. 7c). Genes with a divergent promoter tended to have a peak of accessibility at a greater distance from the start of transcription (in addition to a greater level of conservation further away from the TSS), suggesting there may be a link between promoter accessibility and divergent transcription (Supplementary Fig. 7d–g). We could also observe the presence of nucleosomes starting from both TSSs, alongside increased RNA-seq read density in our exosome mutant datasets (Fig. 6d). Apart from the presence of the TATA box and Inr element at the divergent ncTSS, we could not detect any specific sequence composition unique to these promoters (Supplementary Fig. 7h–i). While there was a higher proportion of correlating versus anti-correlating ncTSS-pcTSS pairs, correlating pairs still represented a

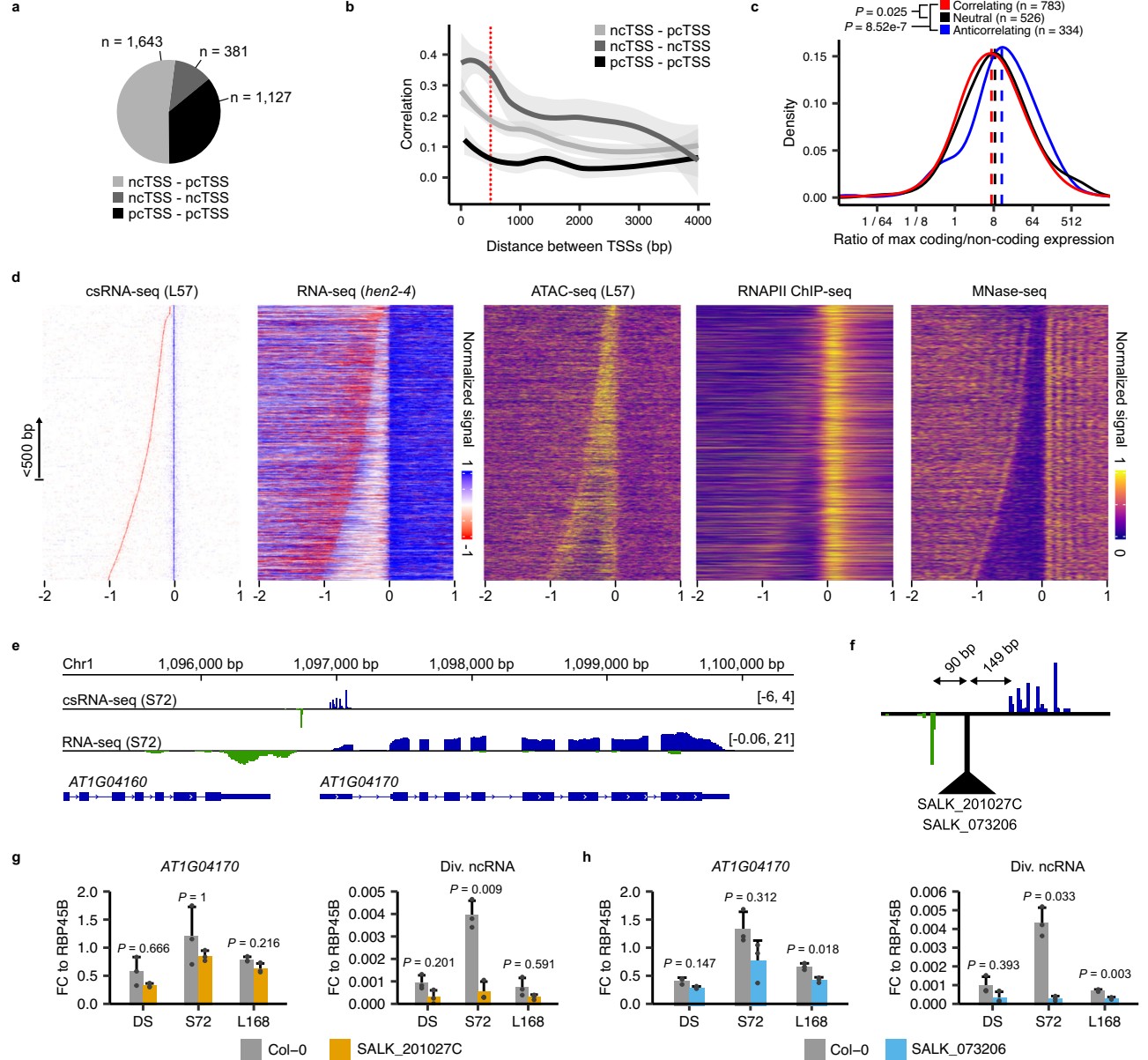

**Fig. 6 | Analysis of bidirectional promoters reveals uncoordinated divergent transcription. a** Pie chart of the number of bidirectional promoters (when two opposite-facing TSSs are present within 500 bp of each other) by annotated TSS type: protein coding TSS (pcTSS) and non-coding TSS (ncTSS). **b** Smoothed Pearson correlation coefficient between bidirectional promoter TSS pairs, plotted against their inter-TSS distances. TSS pairs considered as bidirectional promoters (within 500 bp of each other) are demarcated using a red dashed line. Smoothing was performed using a local polynomial regression fitting. The shaded area represents the 95% confidence interval of the model. **c** Density plot of the ratio (on a log₂ scale) of csRNA-seq signal between each pcTSS-ncTSS pair in divergent promoters, grouped by their Pearson correlation coefficient (anti-correlating: <−0.25; correlating: >0.25). Dashed lines represent the median ratio within each group. Significance testing between correlation groups was performed using two-sided Mann−Whitney tests with Holm correction for multiple testing. **d** Heatmaps of read density from the L57 csRNA-seq, *hen2-4* RNA-seq, L57 ATAC-seq, as well as external RNAPII ChIP-seq from seedlings[110] and MNase-seq from leaves[41] at divergent

promoters, ordered by inter-TSS distances. Data from the csRNA-seq and RNA-seq are row-normalized between −1 and 1, and between 0 and 1 for the ATAC-seq, RNAPII ChIP-seq and MNase-seq datasets. A diagram explaining the process of generating these heatmaps in more detail can be found in Supplementary Fig. 7j. **e** csRNA-seq and RNA-seq read density coverage tracks from the S72 time-point showing the divergent promoter active for the gene *AT1G04170* (units in RPM). **f** Close-up of the csRNA-seq track from (**e**) showing the divergent promoter and the distances between the T-DNA insertion and the TSSs in the SALK_201027C and SALK_073206 mutant lines. **g** RT-qPCR data of the *AT1G04170* mRNA and its divergent ncRNA (Div. ncRNA) in Col-0 and SALK_201027C plants. RNA was extracted for both genotypes from dry seeds (DS), 72 h stratified seeds (S72), and 7 d old seedlings (L168). Data are normalized to the constitutively expressed gene *RBP45B*. Significance testing was performed using two-sided Student's *t*-tests with Bonferroni correction for multiple testing. All experiments were performed with *n* = 3 biological replicates per time-point. Error bars show the standard deviation from the mean. **h** Same as in (**g**), instead comparing Col-0 and SALK_073206 plants.

minority of all discovered cases with most pairs having Pearson correlation coefficients less than 0.25. However the ratio of csRNA-seq signal from each TSS was similar in pairs which both did or did not correlate, with only anti-correlating pairs showing a difference (Fig. 6c). One possible explanation is that the activity of the ncTSS may

be independent from the expression of the pcTSS, and that one of the requirements for divergent transcription may merely be having an accessible promoter. For example, while some genes display clear evidence of divergent transcription correlating with the activity of their primary pcTSSs (e.g. *AT2G29290* and *AT2G29300*; Supplementary

Fig. 8a), others may not (e.g. *LATE ELONGATED HYPOCOTYL 1* (*LHY*); Supplementary Fig. 8b).

As a way to test whether the ncTSS and pcTSS making up a divergent promoter pair are independently transcribed, we selected two genes (*AT1G04170* and *AT3G26650*) demonstrating evidence of divergent transcription for which we could find available mutant T-DNA insertion lines disrupting their promoters (Fig. 6e; Supplementary Fig. 8c). Both T-DNA insertions in the *AT1G04170* promoter in the lines SALK_201027C and SALK_073206 were present near the midpoint of the divergent promoter, which would effectively put enough distance between the two TSSs for them to no longer share a single ACR (Fig. 6f). This gene, as well as its divergent lncRNA, are strongly expressed in the S72 time-point. Using RT-qPCR we observed no effect on the level of transcription of the gene at this time-point, whereas expression of the divergent lncRNA was nearly completely suppressed in both mutant lines, suggesting transcription from the pcTSS is not dependent on the activity of the divergent ncTSS. For *AT3G26650*, the two mutant lines SAIL_1250_D_04 and SALK_138567 have T-DNA insertions within the divergent promoter region and the second exon, respectively (Supplementary Fig. 8d). Showing a peak level of expression at the end of germination, RT-qPCR experiments revealed expression of this gene at this time-point to be nearly completely abolished in both mutant lines whereas the divergent lncRNA was unaffected (Supplementary Fig. 8e, f). These results demonstrate that transcription from one TSS in a divergent promoter can occur even when transcription from the other is disrupted, either when the ncTSS or pcTSS are affected.

### Evidence of transcriptionally active enhancers in plants

We next considered cases of bidirectional non-coding transcription (ncTSS-ncTSS pairs), which as observed previously were detected in both intragenic and intergenic contexts (Fig. 7a–e)[32,48]. The median distance of 211 bp was very close to the 207 bp of divergent promoter pairs (ncTSS-pcTSS), suggesting slightly above ~200 bp to be an optimal inter-TSS distance for shared promoters (Supplementary Fig. 6c; Supplementary Fig. 9a, b). The pattern of conservation at bidirectional non-coding promoters was higher when present within protein coding gene bodies, though in both intragenic and intergenic cases conservation was generally much weaker and less clear than that seen at divergent promoters (Supplementary Fig. 9c, d; Supplementary Fig. 7f, g). In addition, these promoters contained no common sequence base composition found in promoters with the exception of the TATA box and Inr elements (Supplementary Fig. 9e, f). However we could detect clear evidence of RNAPII activity and nucleosome positioning, as well as the active histone marks H3K4me3 and H3K9ac (Fig. 7a; Supplementary Fig. 9g, h). The ratio of csRNA-seq expression from each TSS per bidirectional promoter pair was below 2 in positively correlating cases (which were the majority), as likely these TSSs are being initiated in a more balanced and coordinated manner than that of divergent promoter TSSs (Supplementary Fig. 9i). We experimentally validated the existence of bidirectional non-coding promoters by deleting one such promoter present within an ACR in the body of the gene *SUBTILISIN-LIKE SERINE PROTEASE 2* (*SLP2*) using CRISPR (Fig. 7e; Supplementary Fig. 9j). The *SLP2* ncTSS pair is active in dry seeds. We detected a loss of both sense and antisense transcripts along the *SLP2* locus (corresponding to the two bidirectional non-coding transcripts) in the CRISPR lines (Supplementary Fig. 9k, l) thus confirming the activity of the bidirectional ncTSS pair in dry seeds.

Since bidirectional non-coding transcription is a feature commonly seen within metazoan enhancers, we wondered whether our observed cases could function in a similar role. Previous studies have shown that many non-promoter-associated ACRs in Arabidopsis serve as distal regulators of gene expression[50,51]. While these studies did not observe any transcriptional activity in these regions, we could in fact observe csRNA-seq signal from many of these, such as in the previous

intergenic example in the upstream region of *SPATULA* (*SPT*; for which two previous chromatin interaction dataset have detected loops between the ACR and the *SPT* promoter; Fig. 7b; Supplementary Fig. 10a)[52,53]. Indeed, even in ACRs with no detected TSSs we observed csRNA-seq signal above background levels, suggesting our experiment may be lacking the sensitivity (i.e., read depth) to clearly capture all transcriptional events occurring in our samples (Supplementary Fig. 10b). In order to validate if such regions could be acting as transcriptional enhancers, we cloned the upstream *SPT* ACR (*eSPT*) in a reporter construct upstream of a minimal 35S promoter fused to luciferase (LUC; Supplementary Fig. 10c). After infiltration in *Nicotiana benthamiana* leaves, we observed a median 3-fold increase in LUC signal compared to LUC driven by the mini 35S promoter alone (Supplementary Fig. 10d) thus confirming transcriptionally active ACRs function as enhancers.

We next wished to compile a set of active enhancers during germination from our list of bidirectional non-coding promoters, but wondered whether we could consider additional regions from which only a single TSS was detected, either due to a lack of sensitivity in our csRNA-seq (as an example, the upstream region of *AT1G21000* contains a large ACR with only a single detected TSS but visible RNA-seq signal from both; Supplementary Fig. 10e) or that some enhancers may be only transcriptionally active on a single strand (unidirectional)[54,55]. Expanding this search to intergenic ncTSSs allowed us to generate a set of 1891 putative enhancers, of which most had an overlapping ACR (Supplementary Fig. 10f; Supplementary Data 10). Using the total csRNA-seq signal present over these ACRs as a measure of their activity, we could detect enhancers active over the entire course of germination (Supplementary Fig. 10g). This set of putative enhancers in fact included several previously tested enhancer sequences in Arabidopsis, including 2 / 10 flower and leaf enhancers validated by ref. 51 and 9 / 22 flower enhancers validated by ref. 50, among which the upstream *SPT* enhancer was also shown to be active in floral meristems (Supplementary Fig. 10h, i). Enhancers active in the seedling stage (L57) from our list (defined as the top 500 by total csRNA-seq signal) were enriched for RNAPII and the active histone mark H3K9ac, whereas inactive enhancers (defined as the bottom 500 by total csRNA-seq signal) generally were less accessible and had higher levels of the repressive histone mark H3K27me3 (Fig. 7f). We also found that enhancers were more likely to correlate with the expression of nearby genes (within 5 kb), and genes with a high correlation were enriched for transcription factor activity (Supplementary Fig. 10j, k). These results suggest that csRNA-seq combined with ATAC-seq is an excellent tool for identifying transcriptionally active enhancers in Arabidopsis. Together, our data demonstrate that Arabidopsis enhancer elements are transcribed by RNAPII producing unstable non-coding enhancer RNAs (eRNAs) similar to enhancers in larger plant genomes (such as bread wheat[49]) and metazoan species, thus revealing a previously unknown feature of Arabidopsis enhancer elements.

## Discussion

In this study we used csRNA-seq and ATAC-seq to clarify the role of nascent transcription initiation in the regulation of transcription during the seed-to-seedling transition. First we show that certain processes are actively regulated in dry seeds, including cellular catabolism and ABA-related pathways (Fig. 2; Supplementary Data 4). The expression of ABA-related genes is rapidly and drastically reduced during stratification, with regulation of the transcriptome largely adjusting to a focus on RNA metabolism. During early germination in the light the transcriptome becomes highly enriched for the transcription of ribosome and translation-related genes, although importantly the plant remains susceptible to reactivation of ABA-related genes with accessible promoters likely via residual ABI5 activity. The transition to post-germinative growth is marked by another shift in transcription including to that of DNA replication and protein

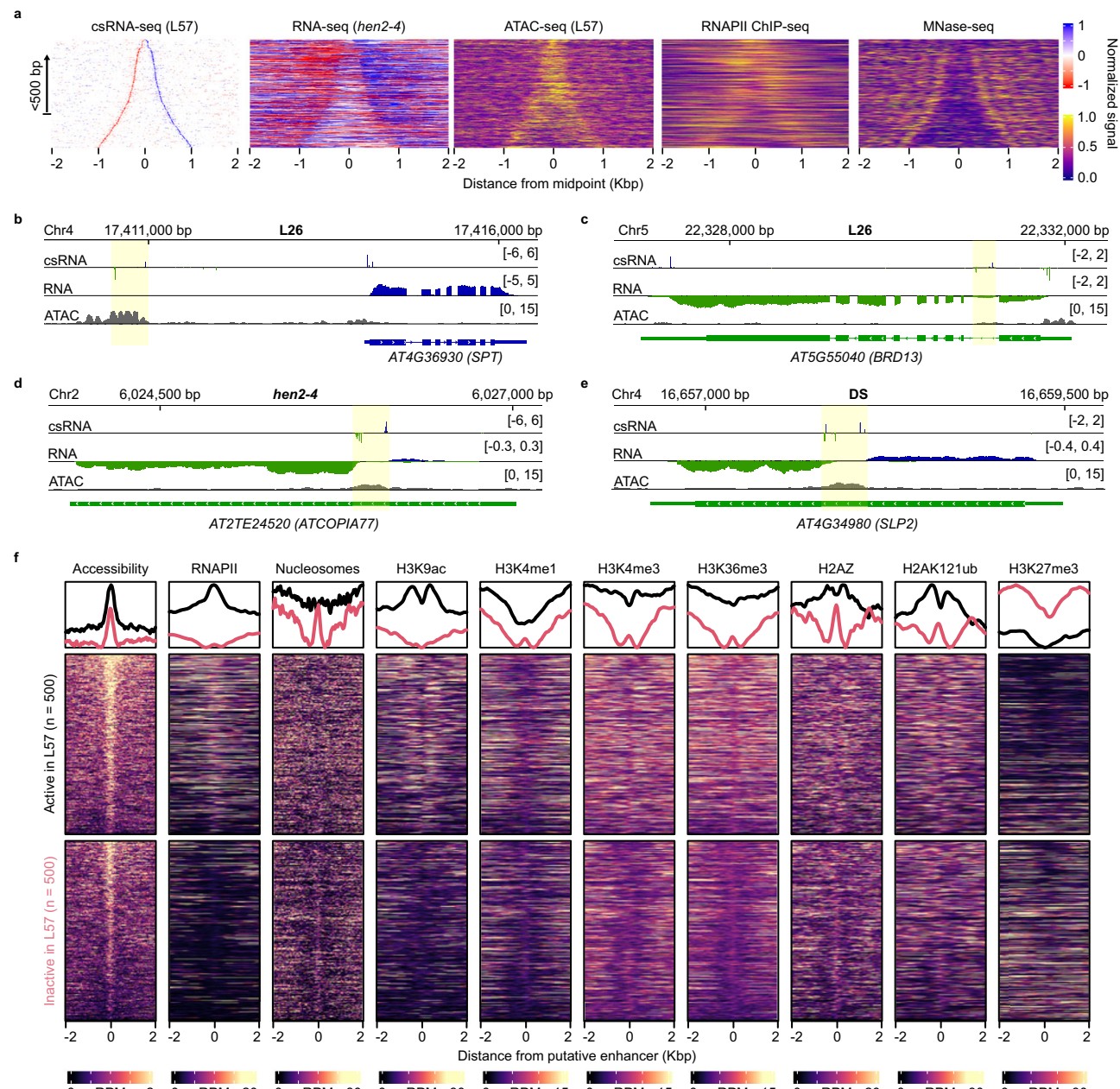

**Fig. 7 | Bidirectional non-coding promoters and transcriptional enhancers during germination. a** Heatmaps of read density from the L57 csRNA-seq, *hen2-4* RNA-seq, L57 ATAC-seq, as well as external RNAPII ChIP-seq from seedlings[110] and MNase-seq from leaves[41] at bidirectional non-coding promoters, ordered by inter-TSS distances. Data from the csRNA-seq and RNA-seq are row-normalized between −1 and 1, and between 0 and 1 for the ATAC-seq, RNAPII ChIP-seq and MNase-seq datasets. These heatmaps were generated in a similar fashion to those found in Fig. 6d, except centering the distance 0 point around the midpoint between the two TSSs. **b** csRNA-seq (top, blue/green), RNA-seq (middle, blue/green) and ATAC-seq (bottom, gray) read density coverage tracks (units in RPM) of the L26 sample showing an intergenic bidirectional non-coding promoter (highlighted in yellow) upstream of the gene *SPT*. **c** Same as (**b**), showing an intragenic bidirectional non-coding promoter present within the intron of the gene *BRD13* in the L26 sample. **d** Same as (**b**), showing a bidirectional non-coding promoter present within a *ATCOPIA77*-family transposable element in the *hen2-4* (csRNA-seq, RNA-seq) and L57 (ATAC-seq) samples. **e** Same as (**b**), showing an intragenic bidirectional non-coding promoter present within the single exon gene *SLP2* in the DS sample. **f** Read density heatmaps of various sequencing datasets for the top 500 and bottom 500 (by total signal intensity) candidate enhancers in the L57 sample. Line plots comparing the relative average for each are shown above. The accessibility heatmap is ATAC-seq read density from the L57 sample. All other datasets are from previously published studies, including the H3K27me3 ChIP-seq[117], nucleosome/MNase-seq[41], H3K9ac ChIP-seq[116], H3K4me1 ChIP-seq[110], H3K4me3 ChIP-seq[115], H3K36me3 ChIP-seq[115], H2AZ ChIP-seq[115], and H2AK121ub ChIP-seq[117] samples.

---

glycosylation genes, and additionally by a general loss in accessibility of ABI5 target promoters (thus signaling a commitment to vegetative growth). These results underscore the importance of nascent transcription during germination: while translation machinery stored within the dry seed is sufficient to translate the necessary proteins for germination[10,11], additional ribosome and translation-related mRNAs must first be nascently transcribed during early germination in order

to allow for sufficient stores of ribosomes for the transition to post-germinative growth. An interesting finding of our work is the detection of csRNA peaks widely distributed in dry seeds. Although a previous report suggested that there may be some level of remaining transcriptional competence[13], metabolically inactive dry seeds may also not provide the necessary microenvironment for many genes to be actively transcribed. Yet, the csRNA-seq data contain clear signatures

of transcription initiation, with corresponding significant levels of RNAPII accumulating at such sites in both dry seeds and during imbibition (see Supplementary Note 2). This observation implies different scenarios, the most likely of them being that csRNA-seq may be detecting RNAPII transcripts initiated from seed maturation and retained at the site of transcription in dormant dry seeds. Alternatively, partially degraded RNAPII transcripts or even some extent of transcriptional elongation taking place in dry seeds, although probably at a very low rate, may be contributing to the csRNA-seq peaks detected in our study. Addressing the functional role of these RNAPII initiated transcripts accumulated in dry seeds as well as their fate upon seed hydration and germination is a very exciting aspect of seed biology that will require future work.

Despite the significance of non-coding transcription (e.g. lncRNAs) in regulating the seed-to-seedling transition[23–25], there has been a lack of experimental data exploring this mode of regulation on a global scale. Using csRNA-seq we successfully identified 8813 unannotated sites of non-coding transcription initiation, and reconstructed putative lncRNAs for 2588 of these (Fig. 4). While our analysis did not uncover any coding potential among the putative lncRNAs (Supplementary Fig. 4j), we did not specifically test whether these sequences encode small peptides which may be of regulatory importance outside the nucleus[56]. Investigating this emerging regulatory aspect of non-coding transcription will be an interesting follow-up to this work. Interestingly we found that non-coding events make up a significant fraction of all detected transcription initiation during the seed-to-seedling transition, ranging from 7 to 17% of all reads found in our samples (Supplementary Figs. 2a, 4a). In addition, of the 19,260 protein coding genes with detectable transcription initiation during the seed-to-seedling transition, 20.7% had antisense transcription ($n = 3985$). We found that antisense transcription proximal to the gene TSS likely has a suppressive effect on gene expression, as such genes accumulated less RNAPII over their gene bodies and did not display typical nucleosome distributions (Fig. 5e; Supplementary Fig. 6e–h). On the other hand, genes with antisense transcription originating nearer to the TTS accumulated the highest average levels of RNAPII over their gene bodies, in addition to having normal nucleosome profiles and higher levels of expression. Our data also suggested that sense and antisense transcription likely does not occur simultaneously as genes with correlated expression of both TSSs were most often expressed in seedlings, the time-point with the highest tissue complexity, and were less expressed than genes with an anticorrelated antisense TSS (i.e., due to the total transcription over the gene being split between the two; Fig. 5d, f, g; Supplementary Fig. 6i). This is in agreement with previous work demonstrating that transcription of *FLC* and its antisense *COOLAIR* are mutually exclusive[45]. One possible explanation for these data is that antisense transcription, when originating near the gene TTS, allows the gene to maintain high levels of RNAPII even when inactive. This could be an additional mode of regulation allowing for highly active genes to be suppressed in a tissue-specific manner as an alternative to changes in chromatin accessibility, histone modifications or DNA methylation. However it is important to note that this may simply be the most common observable form of sense-antisense transcriptional regulation, as examples of positive regulation of sense transcription by the antisense lncRNAs have been reported[57].

Our analysis of promoter structure and dynamics allowed us to characterize the prevalence and role of promoter bidirectionality in Arabidopsis. Although the proportion of bidirectional versus unidirectional promoters can differ between organisms, they have been reported across all kingdoms of life, and are even the most common type of promoter in mammalian genomes[58–66]. Although their existence in plants is disputed, recent studies have suggested they occur for a minority of Arabidopsis promoters[32,47,48]. Our own csRNA-seq experiment indeed confirmed this to be the case: we detected

divergent transcription occurring for nearly 9% ($n = 1643$) of active protein-coding genes ($n = 19,260$; Fig. 6a). Including the additional unique divergent transcription events detected in ref. 32 and ref. 47, the number of protein coding genes described thus far with detectable divergent transcription amounts to 2141 (out of the current 27,533 protein coding genes in the Araport11 annotations[30]), though this is likely an undercount due to the limited number of sampled tissues thus far. Our data suggest that divergent transcription is likely not coordinated between both TSSs, similar to what has been observed in yeast and mouse cell lines[67,68]. Divergent promoters typically were more conserved and had large accessible regions, though the reasons as for why such a low proportion of genes have divergent promoters remains unknown, e.g., compared to 19% in *Escherichia coli*[65], 28% in yeast[69] and over 80% in fungal and mammalian genomes[59].

In addition, we found 381 bidirectional non-coding promoters (Fig. 6a). The expression of the paired TSSs within these types of promoters exhibited significantly higher correlations than those of protein coding gene TSSs found in the same orientation and distance, suggesting some degree of transcriptional coordination may be occurring (Fig. 6b). We successfully validated one of these bidirectional non-coding promoters present within the gene body of *SLP2* (Fig. 7e; Supplementary Fig. 9j–l). We also were able to validate that bidirectional non–coding promoters are acting similarly to transcriptionally active enhancers seen in metazoans, as is the case for the intergenic example found upstream of the gene *SPT* (Fig. 7b; Supplementary Fig. 10a, c, d). As enhancers can sometimes only show evidence of unidirectional transcription[54,55], expanding our list to intergenic regions with such non-coding csRNA-seq signal led us to assemble a final list of 1891 putative enhancers active during germination, which were strongly enriched nearby transcription factors with correlating expression patterns (Supplementary Fig. 10f, g, j, k). Our work has shown that enhancer regions in Arabidopsis do show evidence of transcription as well as bearing active histone marks (Fig. 7f), demonstrating the importance of using techniques with sufficient sensitivity to accurately capture unstable transcriptional activity in the Arabidopsis genome.

## Methods

### Plant materials, growth conditions and genotyping

All *Arabidopsis thaliana* plants used in this study were of the Col-0 accession. The mutant lines *hen2-4* and *rrp4-2* were described previously[28,29]. The mutant T-DNA lines SALK_201027C, SALK_073206, SAIL_1250_D_04 and SALK_138567 were obtained from the Nottingham Arabidopsis Stock Center (NASC). *Arabidopsis thaliana* seeds were surface-sterilized using the gas product of 100 mL of bleach mixed with 3 mL hydrochloric acid for 1 h, followed by stratification in water in the dark at 4 °C. Seeds were grown either in 50 mL liquid ½ MS in 125 mL Erlenmeyer flasks, shaking at 180 RPM, in continuous light at 22 °C or sown in soil and grown in a greenhouse. Germination for each seed batch was monitored using a dissection microscope, scoring for testa rupture, radicle emergence, cotyledon greening, and cotyledon expansion every 2.5 h. For genotyping, DNA was first extracted by grinding a small 5 mm² piece of leaf in 50 μL extraction buffer (EB; 200 mM Tris pH 7.5, 250 mM NaCl, 25 mM EDTA, 0.5% SDS), followed by the addition of 300 μL more EB and vortexed briefly. The homogenate was centrifuged at maximum speed in a benchtop centrifuge at room temperature (RT) for 5 min. 300 μl of the supernatant was transferred to a new tube and gently mixed with 300 μL isopropanol and left to incubate for 5 min at RT. This mix was then centrifuged as before, and the supernatant discarded. The resulting pellet was briefly washed with 75% ethanol by centrifuging at 5000 RPM for 2 min and the supernatant discarded. The tubes were inverted and left to dry overnight before being resuspended in 50 μL TE buffer (10 mM Tris pH 8, 1 mM EDTA). For PCR, 1 μL of resuspended DNA was used in a reaction mix of 10 μL NZYTaq 2X Green Master Mix (NZYTech), 1 μl of

each forward and reverse primer, and 7 μL water. PCR reactions were performed in a thermocycler with an initial denature at 95 °C for 2 min, followed by 35 steps of 30 s denaturation at 94 °C, 30 s annealing at 56 °C, and 2 min extension at 72 °C. A final extension at 72 °C for 2 min was also performed. The PCR products were run for visualization in 1% agarose gels in TAE buffer. Primers used for genotyping each mutant line are described in Supplementary Table 1. Genotyping the *rrp4-2* SNP required an additional restriction enzyme digestion step of the PCR product, which was performed using Eco47I (ThermoFisher Scientific) and following the manufacturer's instructions. The locations of the T-DNA insertions in the NASC lines were confirmed by Sanger sequencing of their genotyping PCR products.

## Total RNA extraction

As the csRNA-seq protocol required high concentrations of pure, good quality RNA for all time-points during germination, we modified an RNA extraction protocol which makes use of a sarkosyl-based extraction buffer optimized for seeds and siliques[70]. Briefly, 50 mg of sample (or 25 mg for dry seeds) was flash-frozen in liquid nitrogen and ground to a powder with a mortar and pestle. 1.5 mL of extraction buffer (EB; 100 mM Tris-HCl pH 9.5, 150 mM NaCl, 5 mM EDTA pH 8, 1% sarkosyl) was prepared per sample, and 7.5 μL beta-mercaptoethanol was added to each EB aliquot and briefly vortexed. After addition of the EB to the frozen ground tissue powder, the homogenate was mixed vigorously and left on ice for 10 min, with intermittent vortexing. This was followed by a 10 min max-speed centrifugation at 4 °C in a benchtop centrifuge. 1 mL of supernatant (avoiding the starchy upper-most layer) was transferred to a new tube and mixed with 1 mL chloroform. Following a 15 s vortexing step, the mixture was centrifuged as before and 900 μL of the upper phase transferred to a new tube. To this was added 90 μL 3 M NaOAc and 1 mL Tris-buffered phenol-chloroform, followed by vortexing for 15 s and another centrifugation as before. As much clear supernatant as possible was moved to a new tube and gently mixed by inversion with an equal volume isopropanol and incubated at 4 °C for 1 h. The precipitated RNA was pelleted by a 20 min max-speed centrifugation at 4 °C. The supernatant was discarded and the pellet washed with 800 μL ice-cold 75% ethanol, followed by a 5 min centrifugation at 7500 × g at 4 °C. The supernatant was discarded, further removed with a pipette after pulse-spinning, and left inverted to dry for 2 min. The pellet was dissolved in 100 μL nuclease-free water, to which 10 μL DNaseI buffer (ThermoFisher Scientific), 5 μL DNaseI (ThermoFisher Scientific), and 1 μL RiboLock (ThermoFisher Scientific) was added and left to incubate at 37 °C for 30 min. The resulting DNA-free RNA was isolated using a modified RNA purification protocol[71]. Briefly, 240 μL nuclease-free water was added to the DNaseI reaction alongside 40 μL solubilization buffer (200 mM Tris-HCl pH 7.5, 10 mM EDTA pH 8, 5% SDS) and vortexed. Afterwards 40 μL 3 M NaOAc and 800 μL Tris-buffered phenol-chloroform was added and again vortexed for 15 s, followed by a max-speed centrifugation for 5 min at 4 °C. The supernatant was transferred to a new tube containing 800 μL chloroform and vortexed for 15 s, followed by another centrifugation as before. The supernatant was again transferred to a new tube, this time to one containing 1 mL ice-cold ethanol, gently mixed by inversion, and left at −80 °C overnight. The following day the tube was centrifuged as before for 20 min. The supernatant was discarded and the pellet was washed with 800 μL 75% ethanol, followed by a 5 min centrifugation at 7500 × g at 4 °C. The supernatant was removed, and again removed using a pipette after pulse spinning. After leaving the tubes to dry inverted for 2 min, the pellet was dissolved in 27 μL nuclease-free water (of which 1.5 μL was used for quantification with a NanoDrop (ThermoFisher Scientific)) and stored at −80 °C. RNA integrity was assessed by running 1 μg RNA in a denaturing RNA gel and 100 ng RNA with a Bioanalyzer RNA analysis chip (Agilent). RNA Integrity Number (RIN) values for our samples

ranged from 7.8 to 9.7, with a median of 8.9 across 16 samples. For the denaturing RNA gel, a gel was prepared using 50 mL MOPS buffer (200 mM MOPS free acid, 50 mM sodium acetate, 0.5 M EDTA pH 8) and 0.7 g agarose powder, which was microwaved until completely dissolved and left to cool. Once cool enough to touch, 600 μL 37% formaldehyde and 5 μL ethidium bromide (10 mg / mL) was added and mixed before casting in a gel tray. Once hardened the gel was left to soak in a gel tank containing MOPS buffer for at least 15 min. RNA was mixed with 11.3 μL sample buffer (6.6 μL formamide, 2 μL 10x MOPS buffer, 2.7 μL 37% formaldehyde), 4 μL tracking dye (0.5% orange G, MOPS buffer, 15% glycerol), and water to a final volume of 20 μL. This mixture was incubated at 65 °C for 5 min and immediately cooled on ice for 1 min. The samples were loaded in the gel and run at 100 V for 45 min before visualization with UV light.

## Small RNA size selection

Selection of small RNAs for the csRNA-seq was performed as described by ref. 26. Briefly, 15 μg of RNA in 15 μL TET buffer (10 mM Tris-HCl pH 7.5, 0.1 mM EDTA, 0.05% Tween-20) was mixed with 15 μL FLB solution (95% formamide, 0.005% bromophenol blue, 0.005% xylene cyanol, 1 mM EDTA) and incubated at 75 °C for 5 min before cooling on ice. This was then run at 200 V for 40 min in a 6 mL 15% urea gel (2.88 g urea, 3 mL 30% acrylamide, 600 μL 10X TBE, 60 μL 10% APS, 2.4 μL TEMED, up to 6 mL with water) which had been pre-run for 30 min in 1X TBE. The gel was incubated in 0.5 μg/mL ethidium bromide (in 1X TBE) for 2 min in the dark with gentle rocking. Using a scalpel and a UV imager, the gel was cut from beneath the lowest visible band to the one-third point from the bottom marker to the top marker for each lane. The gel piece was placed in a 0.5 mL LoBind tube (Eppendorf) with three holes poked out at the bottom using 22 gauge hypodermic needles (Terumo), which itself was placed in a 2 mL LoBind tube (Eppendorf) and centrifuged at maximum speed in a benchtop centrifuge for 2 min. To the resulting slurry 500 μL GEB solution (0.4 M NaOAc pH 5.5, 10 mM Tris pH 7.5, 1 mM EDTA, 0.05% Tween-20) was added and incubated for 2 h in the dark with gentle shaking. This was then transferred to a PVDF 0.45 μm filter column (Merck) held in a 2 mL LoBind tube (Eppendorf) and centrifuged for 2 min at 1000 × g. The column was removed, 1.5 μL GlycoBlue (ThermoFisher Scientific) and 1.5 mL ice-cold ethanol added, and following mixing by inversion the tube was incubated overnight at −80 °C. The following morning the RNA was centrifuged at maximum speed in a cooled benchtop centrifuge (4 °C) for 30 min. The supernatant was removed and the pellet washed with 1 mL ice-cold 75% ethanol, followed by a final centrifugation at 7500 × g for 5 min. The pellet was air-dried for 2 min and resuspended in 6 μL nuclease-free water containing 0.05% Tween-20.

## Preparation and sequencing of csRNA-seq, sRNA-seq and total RNA-seq libraries

Cap-selection of small RNAs and small RNA library preparation were performed as described by ref. 26. Briefly, small RNA samples were incubated at 75 °C for 2 min before cooling on ice. For the sRNA-seq, 0.6 μL of the sample was transferred to a new tube containing 1 μL nuclease-free water containing 0.05% Tween-20 and set aside. 14 μL Terminator mastermix (10.75 μL nuclease-free water containing 0.05% Tween-20, 2 μL Terminator Buffer A (Lucigen), 0.25 μL RiboLock (ThermoFisher Scientific), 1 μL Ter51020 (Lucigen)) was added to the sample and incubated for 1 h at 30 °C. 30 μL CIP mastermix (24 μL nuclease-free water containing 0.05% Tween-20, 5 μL CutSmart buffer (New England Biolabs), 1 μL CIP (New England Biolabs)) was then mixed in and incubated for 45 min at 37 °C. Following this, 50 μL RNAClean XP beads (Beckman Coulter) were added and the solution mixed. 100 μL isopropanol was then added, the tube mixed, and incubated on ice for 10 min. The tube was placed in a magnetic rack for 5 min. The

supernatant was removed and the beads were washed twice using 200 μL 80% ethanol containing 0.05% Tween-20. The beads were then air-dried and resuspended in 25 μL nuclease-free water containing 0.05% Tween-20. The tube was incubated at 75 °C for 3 min before cooling on ice. 25 μL CIP mastermix (18.5 μL nuclease-free water containing 0.05% Tween-20, 5 μL CutSmart buffer [New England Biolabs], 0.5 μL RiboLock (ThermoFisher Scientific), 1 μL CIP (New England Biolabs)) was added, the tube mixed, and the reaction incubated at 37 °C for 45 min. An additional 100 μL nuclease-free water containing 0.05% Tween-20 was added and the tube was placed in a magnetic rack for 5 min. The supernatant was transferred to a new 1.5 mL microcentrifuge tube. 500 μL TRIzol LS solution (Merck) was added and the tube was vortexed. 140 μL chloroform was added and again vortexed. The tube was centrifuged for 10 min in a benchtop centrifuge. The supernatant was transferred to a new 1.5 mL microcentrifuge tube, to which was added 1/10th volume 3 M NaOAc and 0.5 μL GlycoBlue (ThermoFisher Scientific) and vortexed. An equal volume of isopropanol was added and mixed by inversion, and left to incubate on ice for 20 min. The tube was centrifuged at max speed for 30 in in a cooled benchtop centrifuge (4 °C). The supernatant was discarded and the pellet washed with 400 μL 75% ethanol. A small volume containing the pellet was transferred to a PCR tube and spun down before discarding the ethanol wash and leaving to air-dry. The pellet was then dissolved in 3 μL nuclease-free water containing 0.05% Tween-20. This and the previously set aside sample for the sRNA-seq were incubated at 75 °C for 90 s before cooling on ice. To each were added 5 μL decapping mastermix (0.8 μL T4 RNA Ligase buffer (New England Biolabs), 3 μL PEG 8000 (New England Biolabs), 0.3 μL RiboLock (ThermoFisher Scientific), 1 μL RppH (New England Biolabs)) and incubated for 1 h at 37 °C after thorough mixing. The samples were then cooled on ice. sRNA-seq and csRNA-seq libraries were then prepared using the NEBNext Small RNA library kit (New England Biolabs) following the manufacturers protocol and sequenced SE75 in an Illumina NextSeq500. Total RNA was rRNA-depleted and library preparation was performed using the TruSeq Stranded Total RNA library preparation kit with Ribo-Zero (Illumina) following the manufacturer's protocol. Libraries were sequenced PE125 in an Illumina HiSeq2500.

### Seed coat-free embryo sample preparation

Isolation of embryos from seeds was performed as described by ref. 72 with minor modifications. Germinating seeds were placed between two microscope slides with a small drop of water which were rotated and squeezed together with gentle force until the embryos were released from within the seed. The mixture of embryos and seed coats were then kept in a small beaker containing 4 mL ice-cold water until all of the sample had been processed (1–2 g starting material). Following this, 4 mL 80% sucrose was added and mixed gently by inversion. The solution was split into two 5 mL microcentrifuge tubes and centrifuged at 3000 × g for 10 min in a cooled benchtop centrifuge (4 °C). The embryo-containing supernatant was filtered through a piece of miracloth (EMD Millipore), washed with ice-cold water, and briefly dried from below using a piece of paper towel to absorb excess water. The embryos were flash frozen in a mortar and ground to a fine powder with a pestle.

### Nuclear enrichment

Nuclear enrichment was performed as described by ref. 73 with minor modifications. Approximately 500 mg of frozen ground tissue powder was added to 10.5 mL ice-cold nuclear purification buffer (NPB; 20 mM MOPS, 40 mM NaCl, 90 mM KCl, 2 mM EDTA, 0.5 mM EGTA, 0.5 mM spermidine) with added cOmplete Mini EDTA-free protease inhibitor cocktail (Merck) in a chilled mortar and ground with a chilled pestle. The homogenate was collected with a 10 mL serological pipette and passed through a 100 μm cell strainer, as well as a 40 μm cell strainer, into a 15 mL Falcon tube (Corning) kept on ice. The tube was centrifuged at 1200 × g for 10 min in a cooled benchtop centrifuge (4 °C). The supernatant was discarded and the pellet gently resuspended in 1 mL ice-cold nuclear extraction buffer 2 (NEB2; 0.25 M sucrose, 10 mM Tris-HCl pH 8, 10 mM MgCl₂, 1% Triton X-100) with added cOmplete Mini EDTA-free protease inhibitor cocktail (Merck) and transferred to a cooled 1.5 mL microcentrifuge tube. The tube was centrifuged at 12,000 × g for 10 min in a cooled benchtop centrifuge (4 °C). The supernatant was removed and the pellet resuspended in 300 μL ice-cold NEB2. This was then pipetted as a layer above 300 μL of nuclear extraction buffer 3 (NEB3; 1.7 M sucrose, 10 mM Tris-HCl pH 8, 2 mM MgCl₂, 0.15% Triton X-100) with added cOmplete Mini EDTA-free protease inhibitor cocktail (Merck) in a new 1.5 mL microcentrifuge tube. The tube was centrifuged at 16,000 × g for 10 min in a cooled benchtop centrifuge (4 °C). The supernatant was removed, the pellet resuspended in 500 μL ice-cold NPB and transferred to a new 1.5 mL microcentrifuge tube. Nuclei concentration and quality was determined using fluorescence microscopy. Briefly, 1 μL DAPI (0.2 μg/μL) was added to 25 μL nuclei in a 0.5 mL microcentrifuge tube and kept in the dark on ice. A fluorescence microscope was used to count nuclei from 10 μL of the DAPI-stained nuclei inside a Neubauer Improved chamber (Merck).

### Preparation and sequencing of ATAC-seq libraries

Tagmentation and library preparation was performed following the manufacturers instructions in the Diagenode Tagmentation kit (Diagenode). Approximately 50,000 nuclei were transferred to a 1.5 mL microcentrifuge tube and centrifuged at 1000 × g for 10 min in a cooled benchtop centrifuge (4 °C). The nuclei pellet was gently resuspended in 50 μL tagmentation mastermix (25 μL 2X Tagmentation buffer (Diagenode), 16.5 μL PBS pH 7.4, 5 μL nuclease-free water, 0.5 μL 1% digitonin, 0.5 μL 10% Tween-20, 2.5 μL loaded Tagmentase (Diagenode)) and incubated at 37 °C for 30 min, mixing gently every 5 min. The reaction was terminated by the addition of 250 μL Binding Buffer (Diagenode), mixing by pipetting up and down. The mixture was transferred to a Diagenode MicroCHIP Diapure spin column placed in a collection tube (Diagenode) and centrifuged at 10,000 × g for 30 s in a benchtop centrifuge. The flow-through was discarded and 200 μL Wash Buffer (Diagenode) added to the column. The tube was centrifuged as before, flow-through discarded, and the wash step repeated once more. The column was transferred to a 1.5 mL LoBind tube (Eppendorf) and 12 μL DNA Elution Buffer (Diagenode) added directly to the column matrix. The tube was centrifuged once more as before and the final DNA elution stored at −20 °C. For library preparation, 10 μL of sample DNA was mixed with 14 μL of nuclease-free water, 25 μL 2X NEBNext PCR mix (New England Biolabs), and 1 μL of a primer index pair from the 24 UDI For Tagmented Libraries - Set I (Diagenode). A first PCR was performed in a thermocycler using the following conditions: 5 min at 72 °C, 30 s at 98 °C, 5 cycles of 10 s at 98 °C, 30 s at 63 °C, and 1 min at 72 °C. The number of additional required amplification cycles for each sample was determined before continuing. 5 μL of the PCR reaction mix was combined with 5 μL NEBNext PCR mix (New England Biolabs), 3.75 μL nuclease-free water, 1 μL Evagreen dye (Biotium), and 0.25 μL of the previous primer index pair (Diagenode). A qPCR was performed using the following conditions: 30 s at 98 °C, and 20 cycles of 10 s at 98 °C, 30 s at 63 °C, and 1 min at 72 °C. The number of cycles was plotted on the x-axis against the fluorescence readout on the y-axis. The point in the curve at which the reaction reached one-third its maximum fluorescence was found, with the matching cycle number (rounded down to the nearest integer value) used as the final number of additional amplification cycles. This was done by placing the PCR reaction back into the thermocycler, heating to 98 °C for 30 s, followed by the same cycling conditions as before for the determined number of additional cycles. DNA purification was performed by the addition of 67.5 μL AMPure XP beads (Beckman Coulter) to the PCR reaction in a 1.5 mL LoBind tube (Eppendorf) and vortexed for 1 s. After

incubating at RT for 5 min, the tube was placed in a magnetic rack for another 5 min. The supernatant was removed and replaced with 200 μL 80% ethanol, incubating for 30 s. The ethanol wash step was repeated once. The beads were then air-dried for up to 5 min, after which the tube was removed from the magnetic rack and the beads resuspended in 30 μL DNA Elution Buffer (Diagenode) by tapping. This was incubated at RT for 5 min before being placed in the magnetic rack for another 5 min. 28 μL of the elute was transferred to a new 1.5 mL LoBind tube (Eppendorf) and stored at −20 °C. 1 μL was used for quantification and quality control using a Bioanalyzer High Sensitivity DNA chip (Agilent). Libraries were sequenced PE50 in an Illumina HiSeq2500.

## csRNA-seq, sRNA-seq, RNA-seq, and GRO-cap read alignment and processing

GRO-cap raw FASTQ files were downloaded from NCBI SRA using the fasterq-dump program from the SRA toolkit[74] (see Supplementary Table 2). Reads were trimmed for the csRNA-seq, sRNA-seq and GRO-cap using HOMER[75] with the following command: *homerTools trim −3 AGATCGGAAGAGCACACGTCTGAACTCCAGTCAC -mis 2 -minMatchLength 4 -min 20*. For the GRO-seq and total RNA-seq, the same procedure was used with the addition of the *-pe* option. Trimmed reads for all samples were mapped to the TAIR10 genome sequence[76] using STAR[77] with the following options: *−outSAMstrandField intronMotif −outMultimapperOrder Random −outSAMmultNmax 1 −outFilterMultimapNmax 10000 −limitOutSAMoneReadBytes 10000000*. The resulting csRNA-seq and sRNA-seq BAM files were filtered using samtools[78] and custom AWK scripts to exclude reads with more than 1 mismatch, a MAPQ score of less than 10, gaps, soft-clipping or a length outside of the range of 20–70 nucleotides (see Supplementary Table 3 for final read counts per dataset). HOMER format tag directories were created using the makeTagDirectory program[75] with the options *-genome tair10 -checkGC*. Coverage tracks in bedGraph format were generated from the csRNA-seq, sRNA-seq and GRO-cap tag directories using the makeUCSCfile program from HOMER[75], and from the total RNA-seq BAM files using STAR[77]. To obtain coverage tracks from merged replicates, the replicate BAMs were merged with samtools[78] and the same procedure described above used to obtain bedGraph files.

## TSS discovery, quantification, annotation, and creation of normalized coverage tracks

The findcsRNATSS.pl program from the HOMER suite[75] was used to identify TSS peaks from the csRNA-seq samples with the following options: *-includeSingleExons -genome tair10 -ntagThreshold 1 -noFilterRNA -minDistDiff 0.01 -L 1.5*. The corresponding sRNA-seq and total RNA-seq BAM files for each sample were also included in the command via the *-i* and *-rna* options, respectively. Only TSS peaks present in both replicates were kept, and a final merged TSS peak set was created using the mergePeaks program from HOMER[75]. TSSs were annotated as belonging to an existing Araport11 transcript[30] if they were present either within the 5′ region of the transcript or 500 bp upstream to 200 bp downstream of a TSS, or were within the first 25% of the transcript length using R[79] and base Bioconductor packages[80]. TSSs were annotated as protein coding or mRNA TSSs if the corresponding Araport11 annotation was annotated as "protein_coding"; as lncRNA TSSs if the corresponding Araport11 annotation was one of "antisense_long_non_coding_rna", "antisense_rna", "long_noncoding_rna", "novel_transcribed_region", "other_rna", "pseudogene", "transposable_element_gene"; as other ncRNA TSS if the corresponding Araport11 annotation was one of "miRNA", "pre_trna", "small_nuclear_rna", "small_nucleolar_rna"; as putative lncRNA TSSs if there was no matching Araport11 annotation but a putative transcript could be reconstructed; and unstable TSSs otherwise. TSS quantification was performed for the csRNA-seq and GRO-cap tag directories using the

annotatePeaks.pl program from HOMER[75] with the options *-strand + -fragLength 1 -raw*. TMM-normalized counts per million (CPM) from all csRNA-seq samples were obtained by first reading the counts into R[79], then normalizing them using the *calcNormFactors* and *cpm* functions from the edgeR package[81]. The TMM scaling factors were also exported and used for the creation of TMM-normalized bigWig coverage tracks. To do this, the replicate-merged bedGraphs were imported into R[79], normalized using the aforementioned TMM scaling factors, and exported as bigWig files using base Bioconductor packages[80]. TSS peak summits were identified by first merging all csRNA-seq BAM files using samtools[78] and creating a HOMER format tag directory with make-TagDirectory followed by bedGraph coverage files using makeUCSCfile[75]. These were imported into R[79] and the single nucleotide coordinate containing the maximum pileup value identified within each peak using base Bioconductor packages[80], also calculating a TSS width value as the smallest region containing 80% of all read start sites in each TSS peak.

## De novo transcript reconstruction, quantification, and creation of normalized coverage tracks

De novo transcript reconstruction was performed using a combination of two approaches. First, the stringtie program[82] was used with the total RNA-seq BAM files with the following options to generate de novo transcripts: *-m 150 -s 1 -f 0.01*. In addition, the TSS peak summits were used to guide the creation of putative transcripts via the *−ptf* option. The resulting GTF files were merged from all samples and transcripts overlapping existing Araport11 transcripts[30] filtered out using gffcompare[83]. In a second approach, all total RNA-seq BAM files were filtered to remove reads with a MAPQ of less than 10, soft-clipping, or gaps, and then merged into a single file using samtools[78] and custom AWK scripts. A HOMER format tag directory was created using make-TagDirectory, and de novo transcript reconstruction performed using findPeaks with the options *-style groseq -tssSize 100 -minBodySize 100 -endFold 20*[75]. Transcripts overlapping existing Araport11 transcripts[30] were filtered out using gffcompare[83]. The final HOMER and stringtie transcripts were imported into R[79] to generate a unified non-overlapping set of putative transcripts of lengths greater or equal to 200 bp, as well filtering those out whose TSS was not within 50 bp of a TSS peak discovered from the csRNA-seq, using base Bioconductor packages[80]. The final transcript set was obtained by first converting the putative transcript set from GTF to GFF3 format using gffread[83], concatenating with the Araport11 annotations[30] in GFF3 format, and finally sorted using GFF3sort[84]. Transcript quantification was then performed from the total RNA-seq BAM files using stringtie[82], and from the GRO-seq tag directories using the analyzeRepeats.pl program from HOMER[75]. TMM-normalized transcripts per million (TPM) counts and bigWig coverage tracks for the total RNA-seq samples were obtained using the same procedure described above for the csRNA-seq samples.

## ATAC-seq read alignment, processing, and peak annotation

Reads were trimmed using fastp[85] for the adapter sequence CTGTCTCTTATACACATCT. Trimmed reads were mapped to the TAIR10 genome sequence[76] using bowtie2[86] with the following options: *−very-sensitive -X 2000 −dovetail*. Non-nuclear reads, reads overlapping with a manually curated blacklist, and reads with a MAPQ score of less than 10 were discarded using samtools[78]. The remaining reads were deduplicated using Genrich[87] and the accompanying getReads.py script. The filtered BAM was converted into BED format using bedtools[88]. ACR peaks were called using MACS3[89] with the following command: *macs3 callpeak −nomodel −shift −100 −extsize 200 -g 1.191e8 −keep-dup all -p 0.05 -f BED*. To create a final set of peaks from all samples, peaks were first filtered to only keep those with a q value of less than 0.05 and those which were present in both replicates, and finally merged using bedtools[88]. To identify peak summits, reads were resized to 200 bp centered from the start position, merged from all

samples, and converted into bedGraph format using bedtools[88]. This was then read into R[79] and the single nucleotide coordinate containing the maximum pileup value identified within each peak using base Bioconductor packages[80]. To create normalized bigWig tracks for each time-point reads from both replicates were first merged and resized to 50 bp centered from the start position, then reads counted in 10 bp bins along the genome before being converted into bedGraph format with bedtools[88]. This was read into R[79] and the read counts per bin normalized using the *normOffsets* function (with weights for bins outside of peak regions set to 0) and *calculateCPM* function from the csaw package[90] before being exported in bigWig format. To obtain a normalized reads per million (RPM) table of all individual peaks, raw read counts were first obtained using bedtools, read into R[79], and normalized using the *normOffsets* and *calculateCPM* functions from the csaw package[90]. Peaks were annotated based on the distance from their summit to features within the Araport11 annotations[30] using R[79] and base Bioconductor packages[80]. Peaks were annotated in the following order based on the location of their peak summit (with later steps having priority and overwriting an existing annotation for peaks overlapping multiple features): (1) all peaks were initially assigned as "Intergenic"; (2) then peaks overlapping a gene detected in the csRNA-seq were assigned as "Intragenic"; (3) followed by checking if peaks overlapped a transposable element which were assigned as "TE"; (4) and finally peaks overlapping a promoter ($-400$ bp to $+100$ bp) of a detected TSS were assigned as "Promoter".

### ChIP-seq read alignment and processing

ChIP-seq raw FASTQ files were downloaded from NCBI SRA using the fasterq-dump program from the SRA toolkit[74] (see Supplementary Table 2). Reads were trimmed using cutadapt[91] with options *-a AGATCGGAAGAGCACACGTCTGAACTCCAGTCA -A AGATCGGAA-GAGCGTCGTGTAGGGAAAGAGTGT -m 15*. Trimmed reads were mapped onto the genome using bowtie2[86] with option *-X 2000*. The resulting BAM file was used to create a HOMER format tag directory with makeTagDirectory followed by the creation of bedGraph coverage tracks with makeUCSCfile[75]. These were imported into R[79] and exported as bigWig files using base Bioconductor packages[80].

### Differential gene expression/chromatin accessibility analyses and clustering

To determine significantly differentially abundant ACR peaks between samples, the raw counts table (see above) was normalized using the *normOffsets* function from the csaw package[90], followed by the use of the functions *estimateDisp*, *glmQLFit*, and *glmQLFTest* functions from the edgeR package[81]. To determine significantly differentially expressed TSSs and transcripts between samples, the raw counts tables (see above) were normalized using the *calcNormFactors* function, and differential testing performed using the *glmQLFit* and *glmQLFTest* functions from the edgeR package[81]. Principal component analyses were performed using the *prcomp* function from the base stats package in R[79] using normalized counts. For clustering of the ATAC-seq data, the normalized read counts table (see above) was imported into R[79] and ACR peaks with a $\log_2$ variance across all samples of less than 4 filtered out. Then, clustering was performed using the *blockwiseModules* function from the WGCNA package[92] with options *power = 5, TOMType = "signed", networkType = "signed", minModuleSize = 30*. Related clusters were merged using the *mergeCloseModules* function, and clusters with fewer than 1000 members or an average cluster difference between replicates of greater than 25% the standard deviation of all sample averages were removed. For clustering of the csRNA-seq data, the normalized read counts table (see above) was imported into R[79] and clustering performed using the *blockwiseModules* function from the WGCNA package[92] as before with one modified option: *power = 3*. Cluster merging and filtering was performed as described for the ATAC-seq clustering. Transcription factor genes associated with each cluster were identified based on the list of Arabidopsis thaliana transcription factors downloaded from the PlantRegMap database[42]. For clustering of the RNA-seq data, the normalized read counts table (see above) was imported into R[79] and filtered to only keep transcripts where at least 2 individual samples had a TPM of at least 2. Clustering was performed using the *blockwiseModules* function from the WGCNA package[92] as before with one modified option: *power = 6*. Cluster merging and filtering was performed as described for the ATAC-seq clustering. Gene ontology enrichment was performed for all clusters using the gprofiler2 package[93], and plotted as word clouds using the *anno_word_cloud_from_GO* function from the simplifyEnrichment package[94] alongside Z-score heatmaps of chromatin accessibility or gene expression data created using the ComplexHeatmap package[95]. A set of constitutively expressed TSSs were selected by sorting all TSSs in ascending order by their coefficient of variation across all samples, removing those with a minimum csRNA-seq expression less than 50 CPM, and keeping the top 500.

### De novo motif enrichment and analyses

FASTA files of the ACR peaks were generated by first adjusting the peaks to all be sized 500 bp (centered around the peak summits) and then using bedtools[88] to extract their corresponding sequences from the TAIR10 genome sequence[76]. A similar approach was used to generate FASTA files of the csRNA-seq TSS promoters (from 400 bp upstream to 100 downstream of each TSS peak summit). The streme program[96] was used to find de novo motifs from each ATAC-seq and csRNA-seq cluster individually, using all ACR and TSS peaks as the background sequences, respectively. The resulting motifs were imported into R[79], then clusters of similar motifs were identified using a target overlap coefficient of greater or equal to 0.9 and merged using the *merge_motifs* function and plotted using the *view_motifs* function from the universalmotif package[97]. The enrichment of each motif within the cluster sequences was determined using centrimo[98] for positional enrichment and sea[99] for overrepresentation enrichment (as compared to the background, i.e., all ACR peaks or promoters). Motifs with an overrepresentation enrichment $q$ value greater than 0.01 were discarded. To identify putative transcription factors associated with each motif, Tomtom[100] was used to determine similarity to known *Arabidopsis thaliana* motifs downloaded from PlantTFDB[101]. Hits with a $p$ value greater than 0.05 were discarded. A single representative transcription factor was manually assigned based on whether the transcription factor belonged to a cluster in which the motif had the most significant enrichment, and whether the transcription was highly expressed.

### Identification of exosome-sensitive non-coding transcripts and TSSs

To analyze the impact of exosome deficiency on transcription initiation and the transcriptome, we examined the $\log_2$ fold changes of all TSSs not associated with a protein coding gene (i.e., non-coding TSSs) in the csRNA-seq as well as non-coding transcripts in the total RNA-seq between the wild-type L57 sample and the *hen2-4* and *rrp4-2* mutant samples (see above). Non-coding TSSs and transcripts were marked as significantly differentially expressed if their absolute $\log_2$ fold change was greater or equal to 2, and their $q$ value was less than 0.05. Within the csRNA-seq and total RNA-seq, TSSs or transcripts which were significantly upregulated in the mutants were classified as exosome-sensitive, and those with an absolute $\log_2$ fold change less than 1, a $q$ value greater than 0.05, and a maximum CPM expression value greater than 1 in the csRNA-seq or a maximum TPM expression value greater than 0.1 in the total RNA-seq were classified as exosome-insensitive. Hypergeometric tests were used to determine whether there was significant overlap in the presence of exosome-sensitive TSSs and transcripts between the mutants via the *phyper* function from the stats package in R[79]. The GC content of non-coding TSSs and transcripts was

calculated in R[79] as the moving average GC proportion across all tested features in windows of 20 in a 2 kbp region centered around the TSS using the zoo package[102] and base Bioconductor packages[80]. Library GC content plots were created using the GC content probability density function output of the makeTagDirectory program from HOMER run using the -checkGC option[75].

## Normalized read coverage heatmaps

Read densities in 3 or 4 Kbp windows around TSS peak summits were first obtained from normalized bigWig files (see above) in R[79] using the ScoreMatrix function from the genomation package[103]. Then, for each window the scores were trimmed to their 90th percentile values and rescaled between 0 and 1. For stranded data this was performed individually for windows specific to each strand, and a final set of values per window was calculated by subtracting the rescaled antisense strand values from the rescaled sense strand values. The final values were plotted using the ComplexHeatmap package[95]

## Promoter characteristics of csRNA-seq TSSs

To plot the nucleotide composition of the genomic regions surrounding the start site of mapped csRNA-seq reads, the filtered mapped BAM files (see above) were converted to BED format, resized from 50 bp upstream of the read start to 10 bp downstream, and the corresponding TAIR10 genome sequence[76] extracted using bedtools[88]. The resulting FASTA files were read into R[79] using base Bioconductor packages[80], converted to information content matrices using the create_motif function and plotted using the view_motifs function from the universalmotif package[97]. The two nucleotide Inr element preferences were found by obtaining the first nucleotide upstream of each TSS peak summit (above) as well as the nucleotide of the summit itself using R[79] and base Bioconductor packages[80]. Promoter base composition plots were created by first obtaining the genomic sequences surrounding each TSS peak summit (either 200 bp or 1000 bp) and then calculating the total proportion of each nucleotide using R[79] and base Bioconductor packages[80]. The distance between csRNA-seq TSS peaks and the annotated TSS coordinate from Araport11[30] was calculated as the closest distance between the peak region and the 1 bp annotated TSS coordinate, with a distance of 0 assigned when they overlapped. PhyloP and PhastCons conservation scores for Arabidopsis thaliana calculated from 63 plants were downloaded from the PlantRegMap database[42] and used to plot the average sequence conservation of the region surrounding each TSS in R[79] via the EnrichedHeatmap package[104]. Processed MNase-seq data representing nucleosome position along the genome in leaf tissue was downloaded from the PlantDHS database[41] and analyzed in the same method described for the sequence conservation. Average ATAC-seq read density at TSS promoters was obtained using the EnrichedHeatmap package[104] for different expression brackets of csRNA-seq CPM expression: 0–10, 10–100, 100–1000, and >1000.

## Annotation and analysis of antisense TSSs

The annotated csRNA-seq TSSs were loaded into R[79] and non-coding TSSs which overlapped the antisense strand of a protein coding gene (or 200 bp downstream of the TTS) from the Araport11 annotations[30] with a detectable TSS were assigned as antisense to the protein coding TSS. For protein coding genes with multiple TSSs, the highest expressing one was selected as the single representative. Antisense TSSs were classified as "proximal" if they were present within the first 50% of the protein coding gene body and were further than 1 Kbp from the TTS. The remaining were classified as "distal". The ratio of sense to antisense expression was calculated by taking the maximum csRNA-seq CPM expression of each TSS across all samples (excluding the exosome mutants) and plotted as a density function on a $\log_{10}$ scale using the density function from the stats package in R[79]. Similarly, the

Pearson correlation coefficient between sense and antisense TSS was calculated with the cor function from the stats package in R[79] and plotted against the maximum csRNA-seq CPM ($\log_2$ transformed) expression of the protein coding TSS using the geom_smooth function from the ggplot2 package[105] using data from all samples (excluding the exosome mutants). Average TSS conservation, ATAC-seq read density, MNase-seq read density, and RNAPII ChIP-seq read density plots were generated as described above (see Supplementary Table 2). The expected versus observed number of antisense TSSs per cluster was performed by comparing the expected number for each cluster (calculated as the fraction of clustered TSSs being antisense multiplied by the cluster size) with the observed number of antisense TSSs per cluster using chi-square tests via the chisq.test function from the stats package in R[79]. Correlating and anti-correlating sense and antisense pairs were plotted as Z-scores using the ComplexHeatmap package[95]. Promoter nucleotide composition plots were created as described above.

## Annotation and analysis of bidirectional promoters

Bidirectional promoters were identified using R[79] and base Bioconductor packages[80]. Briefly, opposite facing TSS pairs were classified as either (1) protein coding / protein coding, (2) non-coding / protein coding, or (3) non-coding / non-coding. Then, the Pearson correlation coefficients between all pairs were calculated using the cor function from the stats package in R[79] and plotted against their inter-TSS distance using the geom_smooth function from the ggplot2 package[105]. Only TSS pairs with an inter-TSS distance less than or equal to 500 bp were considered true bidirectional promoters. The ratio of maximum expression between TSS pairs was calculated by dividing the maximum csRNA-seq CPM value of the higher expressing TSS within each pair by the maximum csRNA-seq CPM value of the lower expressing TSS. Average TSS conservation, ATAC-seq read density, MNase-seq read density, RNAPII ChIP-seq read density, H3K4me3 ChIP-seq and H3K9ac read density plots were generated as described above (see Supplementary Table 2). ReMap transcription factor density plots were obtained by first downloading the ReMap 2022 non-redundant transcription factor peaks in BED format[106], converting to bedGraph format using bedtools[88], then the average peak coverage calculated using the normalizeToMatrix function from the EnrichedHeatmap package[104] in R[79].

## Annotation and analysis of putative enhancer regions

Putative enhancers were initially selected from all non-coding bidirectional promoters and intergenic non-coding unidirectional promoters. They were then resized to have a width of at least 500 bp. Those from intragenic non-coding bidirectional promoters were manually curated to prevent overlap with the protein coding gene promoter. Enhancer activity across the time-series was calculated by the sum of all csRNA-seq signal over the region, which was obtained from the TMM-normalized bigWig files for each sample (see above) using the bigWigAverageOverBed program from the UCSC genome browser tools[107]. The Pearson correlation coefficient between these scores and the csRNA-seq CPM expression values of protein coding TSSs was calculated using the cor function from the stats package in R[79]. A random distribution of correlations was obtained using inter-chromosomal enhancer−protein coding TSS pairs, which was compared to the distribution of intra-chromosomal enhancer−protein coding TSS pairs within 5 Kbp of each other using the density function from the stats package in R[79]. Protein coding TSSs with a correlation value of 0.5 or greater to a nearby enhancer (within 5 Kbp) were used for gene ontology enrichment using the gprofiler2 package[93]. ATAC-seq read density, MNase-seq read density and external ChIP-seq read density plots were created as described above (see Supplementary Table 2).

## cDNA preparation and RT-qPCR

1 μg of RNA in 7 μL nuclease-free water was mixed with 3 μL DNaseI mastermix (1 μL 10X DNaseI buffer (ThermoFisher Scientific), 1 μL 25 mM $MgCl_2$, 1 μL 1 U/μL DNaseI [ThermoFisher Scientific]) and incubated at 37 °C for 30 min before being inactivated by the addition of 1 μL 50 mM EDTA and incubating at 65 °C for 10 min. Following this gene and strand-specific reverse transcription (RT) was performed by the addition of 19 μL RT mastermix (1 μL 10 mM dNTP, 0.2 μL of each 10 μM primer, 4 μL 5X Maxima RT buffer (Thermo-Fisher Scientific), 0.2 μL Maxima RT (ThermoFisher Scientific), nuclease-free water up to 19 μL) and incubating for 30 min at 50 °C, with a final inactivation step at 85 °C for 5 min and diluting the final cDNA to 200 μL with nuclease-free water. For RT-qPCR, 3 μL of cDNA was mixed with 1.5 μL nuclease-free water, 0.3 μL of forward and reverse 10 μM primer, and 5 μL 2X SYBR Green mastermix (Thermo-Fisher Scientific). Fluorescence was monitored during each amplification step using the following cycling conditions: initial denaturation at 95 °C for 10 s, followed by 45 cycles at 95 °C for 10 s, 60 °C for 30 s, and 72 °C for 30 s. Relative expression was determined by calculating the delta $C_t$ between each target gene and the control gene (*RBP45B*, which is constitutively expressed during the seed-to-seedling transition). See Supplementary Table 1 for a complete list of primers used in this study.

## smFISH

smFISH method in root squashes of 7-day-old seedlings was performed as described by ref. 108. For 1 h imbibition and 6 h light samples, modifications were made as follows. Embryos were isolated by gently pressing seeds between a microscope slide and cover glass and quickly placed onto a drop of 4% formaldehyde. Embryos were fixed for 30 min and then washed three times in 1× PBS. Samples were then squashed between a microscope slide and cover glass to obtain monolayers of cells. Tissue permeabilization and clearing was achieved by air-drying the slides at room temperature for 1 h and then immersing them at 4 °C in 70% ethanol for 30 min, followed by 30 min in 100% methanol, and overnight in 70% ethanol. The ethanol was then left to evaporate at room temperature before two washes were performed with wash buffer (containing 10% formamide and 2X SSC). The hybridization steps, mounting and RNAse control were performed as described by ref. 108. Samples were imaged on a Zeiss LSM800 inverted confocal microscope equipped with a cooled quad-port CCD (charge-coupled device) ZEISS Axiocam 503 mono camera, using a 63X water-immersion objective (1.20 NA). A series of optical sections with z-steps of 0.22 μm were collected. The following wavelengths were used for fluorescence detection: for smFISH probes labeled with Quasar670 an excitation filter 625–655 nm was used and signal was detected at 665–715 nm; for DAPI an excitation filter 335–383 nm was used, and signal was detected at 420–470 nm. smFISH probes were designed using the LGC Biosearch Technologies' Stellaris® version 4.2. The sequences of the probe used in this study are shown in Supplementary Table 1.

## CRISPR-Cas9 mutant generation

Arabidopsis *CRISPR* lines were generated following a protocol described previously[109]. The Golden Gate method was used to construct a vector expressing two guide RNAs (gRNAs) that target *SLP2* enhancer region to generate deletions. gRNAs were designed using the following web tool: http://crispr.hzau.edu.cn/cgi-bin/CRISPR2/CRISPR. For assembly, the two gRNA sequences were incorporated into PCR forward and reverse primers (Supplementary Table 1). The PCR fragment was amplified from pCBC-DT1T2, purified and inserted into pHEE401E binary vector (carrying *CAS9* construct) by Golden Gate reaction. The binary vector obtained was transformed into *A. tumefaciens* GV3010 strain and subsequently transformed into Arabidopsis Col-0 by the floral dip method.

## LUC activity assays in *N. benthamiana* leaves

Plasmids for LUC assay were constructed using pGreen II 0800 LUC (kindly provided by Soraya Pelaz) that carries Firefly *LUCIFERASE* coding sequence without a promoter and a *35S:RENILLA* construct. To generate the control plasmid, we first inserted the mini35S promoter sequence upstream of the *LUCIFERASE* coding sequence using BamHI and NotI (New England Biolabs) restriction enzymes to obtain the pGreen II 0800 mini35S:LUC control vector. Next, we amplified by PCR the *SPT* enhancer regions (*eSPT*), and inserted the purified fragment upstream of the mini35S in the pGreen II 0800 mini35S:LUC plasmid by restriction/ligation with KpnI and SalI (New England Biolabs). All primers are listed in Supplementary Table 1. pGreen II 0800 mini35S:-LUC/eSPT_mini35S:LUC plasmids were transformed by electroporation into *A. tumefaciens* strain GV3101 (harboring pSOUP; a gift from Ignacio Rubio-Somoza). Agrobacterium cultures were prepared by inoculating a single colony, and grown at 28 °C in a shaking incubator. Cultures were harvested by centrifugation at 4 °C and 2000 × *g* for 10 min. Cell pellets were resuspended in a solution containing 10 mM 2-(*N*-morpholino)ethanesulfonic acid (MES), 10 mM MgCl2 and acetosyringone (150 μM). The OD600 adjusted to 0.8. *Nicotiana benthamiana* (*rdr6i*) plants were grown for approximately 3–4 weeks prior to agroinfiltration. Expanded leaves were infiltrated by applying pressure on the abaxial surface of the leaf with a disposable 1 mL syringe containing the Agrobacterium suspension. Each leaf was infiltrated with all three constructs. Each infiltration zone was approximately 2 cm in diameter. Agroinfiltrated plants were incubated for 42 h. Square sections within the infiltration zone were cut using a razor blade, harvested individually in 1.5 mL tubes, snap-frozen in liquid nitrogen, and stored at −80 °C prior to measurement in dual LUC assays. Dual luciferase (LUC) assay extracts were prepared using the Dual-Luciferase Reporter Assay System kit (E1910, Promega). *N. benthamiana* leaf samples were ground to powder and resuspended in 200 μL of 1× passive lysis buffer (PLB) provided in the Dual-Luciferase Reporter Assay System kit. The cellular debris was pelleted by centrifugation at 7500 × *g* for 5 min. The assay was performed using a microplate luminometer (Berthold). 7 μL of the supernatant was loaded into a well of a white flat bottom 96 well plate containing 35 μL of luciferase assay reagent. Following the first luciferase (Firefly) read, 35 μL of Stop &Glo reagent was added to each well to perform a second read (Renilla). Luciferase assay reagent and Stop&Glo reagent components are provided in the Dual-Luciferase Reporter Assay System kit.

## ChIP-qPCR

RNAPII ChIP of dry and 72 h stratified seeds were performed as described in ref. 111 with minor modifications. Briefly, 10 mg of sample was flash frozen with liquid nitrogen and ground to a fine powder. The powder was then crosslinked with 1% formaldehyde in PBS for 15 min followed by the addition of glycine to quench the remaining formaldehyde for 10 min. The powder was collected at the bottom of the tube via centrifugation, washed by resuspending in cold PBS, then centrifuged again. Nuclear extraction was performed using Honda buffer (20 mM HEPES, 0.44 M sucrose, 1.25% Ficoll, 2.5% Dextran T40, 10 mM $MgCl_2$, 0.5% Triton X-100, 5 mM DTT, 1X EDTA-free protease inhibitor tablet (Roche)). After centrifugation nuclear pellets were resuspended in low salt (LS) buffer (50 mM Tris-HCl pH 8, 150 mM NaCl, 1 mM EDTA, 0.1% SDS, 1% Triton X-100, 0.1% Na-deoxycholate, 1X EDTA-free protease inhibitor tablet (Roche)) and sonicated using a Diagenode Bioruptor set to Low Power, 30 s on/30 s off, for 15 cycles. Sonicated samples were pre-cleared for 1 h using 15 μL Dynabeads Protein G (Invitrogen), followed by immunoprecipitation with 1 μg anti-RNAPII (C15200004, Diagenode) bound to 15 μL Dynabeads Protein G (Invitrogen) for 4 h. 1% of the sonicated solution was taken before the IP as input. The beads were then washed once LS buffer, once with high salt (HS) wash buffer (500 mM NaCl, 20 mM Tri-HCl pH 7.5, 2.5 mM

EDTA, 0.05% Na-deoxycholate, 1% Triton X-100, 1X EDTA-free protease inhibitor tablet (Roche)), once with LiCl wash buffer (250 mM LiCl, 20 mM Tris-HCl pH 7.5, 2.5 mM EDTA, 0.5% Na-deoxycholate, 0.5% NP-40, 1X EDTA-free protease inhibitor tablet (Roche)), and once with TE buffer (10 mM Tris-HCl pH 7.5, 1 mM EDTA). DNA was eluted and reverse-crosslinked overnight from the beads and input solutions at 65 °C in elution buffer (10 mM Tris-HCl pH 8, 0.3 M NaCl, 5 mM EDTA, 0.5% SDS), then treated with RNase A for 30 min at 37 °C, proteinase K for 1 h at 55 °C, and finally extracted with phenol-chloroform. qPCR was performed using primers against *ACTIN7*, *DOG1*, and the negative control region *IGN5*[111,112]. Primer sequences can be found in Supplementary Table 1.

## Reporting summary
Further information on research design is available in the Nature Portfolio Reporting Summary linked to this article.

## Data availability
All csRNA-seq, sRNA-seq, RNA-seq and ATAC-seq datasets generated in this study are available from the NCBI GEO repository under the accession GSE250331. Plasmids and mutant lines generated in this study are available upon reasonable request to the corresponding author. Source data are provided with this paper.

## Code availability
Original code and data can be accessed from https://github.com/bjmt/Tremblay_et_al_2024_Seed_to_seedling.

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

## Acknowledgements

The authors thank Heike Lange and Peter Brodersen for providing the *hen2-4* and the *rrp4-2* mutant seeds, respectively, and Ignacio Rubio-Somoza for providing pSOUP Agrobacterium strain. We are grateful to Paloma Más for critically reading the manuscript. This work was supported by the Junior Leader Fellowship [LCF/BQ/ PI19/11690003], from "laCaixa" Foundation [ID100010434] awarded to J.I.Q., the Severo

Ochoa Excellence Programme for Centres 2016-2019 (SEV-2015-0533) awarded to CRAG (MICIN/AEI/10.13039/501100011033), and the EUR2021-122003 and PID2019-110510GA-I00 grants (MCIN/AEI) awarded to J.I.Q. B.J.M.T. holds an FPI predoctoral fellowship (PRE2019-090156) from MICIN/AEI. Y.C. is supported by a fellowship from the Chinese Scholarship Council.

## Author contributions

J.I.Q. conceptualized and supervised the study and acquired funding. B.J.M.T. and J.I.Q. designed the experiments, performed the cloning and LUC assay experiments of putative enhancers, and wrote the manuscript with input from all authors. B.J.M.T. performed the sequencing and ChIP-qPCR experiments, designed and performed the computational analyses and assembled the figures. X.Z. and S.R. performed the smFISH experiment. J.I.Q. and Y.C. generated the CRISPR-Cas9 mutants. C.P.S. performed RT-qPCR experiments of the T-DNA and CRISPR-Cas9 mutants.

## Competing interests

The authors declare no competing interests.
