## [Peer Review File · Nature Communications]

Interplay between coding and non-coding regulation drives the Arabidopsis seed-to-seedling transitionReviewer #1 (Remarks to the Author):

In this report, the authors used capped small RNA-seq (csRNA-seq), sRNA-seq, and total RNA-seq to identify and quantify genes/regions undergoing active transcription initiation in dry seeds and at different stages of germination. They also used ATAC-seq to analyze chromatin accessibility. In addition, the data were used to analyse TSS and other cis-elements. The study identified large sets of genes that were under active transcription, even in dry seeds.

The authors conducted careful and comprehensive analysis of the various transcriptomics data and ATAC-seq data. However, my main concern is the method used to identify newly initiated Pol II transcripts, which is flawed. This method compares csRNA-seq reads with sRNA-seq reads to identify enriched reads in the csRNA-seq data set as newly initiated Pol II transcripts. However, this method is prone to generating false positives, as the enriched reads may actually be cap-containing degradation products of steady-state RNAs. For instance, let us assume that all small RNAs in an RNA sample are degradation products of steady-state RNAs, with some of them being cap-containing fragments from the 5' region of RNAs. In sRNA-seq, these small RNAs would be all identified, and during csRNA-seq library construction, the cap-containing fragments would be enriched. By comparing sRNA-seq reads with csRNA-seq reads, the enriched reads would be identified as 'newly initiated Pol II transcripts,' although they are just degradation products of steady-state RNAs.

This method's flaw could be the reason why the authors identified a large number of genes undergoing active transcription initiation in dormant dry seeds, while the enriched cap-containing RNAs were possibly degradation products of stored mRNAs in the dry seeds. Dormant dry seeds are metabolically inactive organisms that do not provide the necessary microenvironment for many genes to be actively transcribed. If many genes undergo active transcription in dormant seeds, it would lead to significant loss of seed reserves, as cells would need to produce nucleotides, as well as proteins and enzymes to synthesize RNA building blocks and for the transcriptional process, severely damaging the seeds' function.

I understand that the csRNA-seq method was published in Genome Research by another group four years ago and that the authors of this manuscript have invested a lot of effort in conducting the experiments and analyzing the data. However, I would reconsider my view on the report if the authors can show that my concerns about the method are unfounded.

Reviewer #2 (Remarks to the Author):

This manuscript by Tremblay and colleagues describes an interesting and important study into nascent transcription during germination. The context of the study - that we currently understand little about how and when transcription is initiated during germination - is a significant unknown in the field of germination biology. To address this the authors apply capped small (cs) RNA-seq, a method that identifies RNAs at a very early stage of transcription (2-60 nt) and consequently allows transcriptional start sites to be mapped. In parallel the authors generate accessible chromatin maps and traditional RNA-seq. All of these data are generated across a time series of germination from dry seeds to young seedlings. Through a series of clever analyses the authors make a number of observations that substantially advance our understanding of seed germination. These include detecting active transcription in dry seeds - essentially prior to germination - and describing genome-wide transcription of many long-non-coding RNAs in germinating seeds. They illustrate how nascent transcription and chromatin accessibility data can be used to define gene regulatory programs active during germination and identify candidate transcription factors. This analysis then allows them to define more clearly the molecular mechanism for ABA sensitivity in seeds that have commenced germination, which is due to continued promoter accessibility in ABA response genes after their transcription has begun to decrease. The authors also investigate properties of antisense transcripts in germination, deducing that antisense and sense transcripts from individual genes are likely to exist in different cell types because the sense transcript appears to disrupt transcription of the antisense transcript. Bidirectional promoters are another area of focus, where the authors use their data to create an extensive resource of promoters driving different combinations coding and non-coding loci and to understand where these are or are not co-ordinately expressed. The study of bidirectional promoters is notable in that there is controversy in the field about their existence; however, I did at times struggle to follow this section

of the manuscript and have some suggested revisions to aid clarity.

This work substantially advances our knowledge of the processes regulating gene expression in germinating seeds, and has broad/general implications for gene regulation in Arabidopsis. Prior to this study, we did not know how early transcription was initiated during germination, nor how extensive non-coding transcription was. Overall I am very impressed and feel the manuscript merits publication in Nature Communications with some minor revisions, details of which follow. These revisions should only require work on the manuscript text and possibly some minor reanalyses of existing data.

Signed, Mat Lewsey.

Suggested revisions and points for clarification

- Fig. 1c - in legend clarify if genes are ordered the same in each panel.
- Fig. 1d - what is the interpretation of the foci of unspliced transcript outside of the nucleus?
- Ln 196. 'however none of the TFs belonging to the C4 cluster had similar binding sites (Dataset S7). ' I got lost here. Do you mean expressed in rather than belonging to?
- Ln 226. 'as a nucleosome can be observed in the promoters of DS-specific genes and more generally at instances of the motif itself'. Expand on this to be more clear about what you think that nucleosome indicates/is doing.
- Ln 296. 'the presence of a proximal antisense TSS generally led to lower levels of expression'. Correlated with, not led to - causality is not demonstrated here.
- Ln 312. 'these data suggest that sense and antisense transcription likely cannot occur simultaneously and only appear to correlate in our bulk sequencing data as a result of their expression in different tissues.' If this is the case, what do you propose is the function of the antisense transcripts? Previously I thought they were assumed to act in feedback regulation of the sense transcript within the same cell.
- Fig 6d. Explain in left most panel, cs-RNA-seq, what the red and blue vertical lines indicate.
- Ln 347. I got quite confused in this section. 'While there was a higher proportion of correlating ncTSS-pcTSS pairs than not, these still represented a minority of all discovered cases.' I'm uncertain what you mean here - please rephrase. In the relevant figure, 6d, why has row normalization been applied non-uniformly within rows (eg. csRNA-seq, legend says normalization was applied from 1- to 1, when scale is -2 to 1, which will have differencing effects on interpretation of every row in the plot since the TSSs are at different points relative to this scale)? What is the value of n in fig 6d? It appears from fig 6d that every nc/pc pair is expressed in an anti-correlated manner, but that's not what is described in the main text. Is that what you mean by divergent and somehow you have selected for these cases in your analysis? If not, please define divergent. And why is the hen2-4 data presented for RNA-seq - please explain the significance of that mutant here.
- More broadly, I think the importance / biological relevance of the exosome mutants has not been explained in enough detail for readers from outside the non coding RNA discipline.
- Fig 7a. As with 6d, what do the red and blue vertical lines indicate in the csRNA-seq panel? If I apply the scale bar from the right to them, it suggests to me that all these sites are anticorrelated. I don't understand the non-uniform row normalization within rows here nor the particular relevance of the hen2-4 mutant here either.
- Fig 7b-d. Label the rows directly in your genome browser screenshots. 7f, states top and bottom 500 candidate enhancers in the legend, but active/inactive in the figure itself; are these the same

thing, and how were inactive enhancers defined? This point applies to the related main text also.

- Supplemental information - detection of exosome sensitive transcripts is described here, but the significance is not explained in enough detail. Please expand on what this tells you.

- This may demonstrate the limits of my understanding of nascent transcription profiling, but how do you tell whether this is active transcription or paused/poised polII at genes that will become transcribed later? Put another way, considering all of the 'active' TSSs you detected in dry seeds, how many had a clear RNA-seq signal (ie. full length transcript) at that same time point? Fig S1 provides a nice single example of where csRNA-seq and RNA-seq match up in the dry seed, but was this how it appeared at all csRNA-seq sites? What I am getting at is could polII be recruited to those promoters during late seed development and initiate transcription, then pause and sit there until germination, as opposed to your results showing active transcription in the dry seed, or can you rule that out?

RESPONSES TO REVIEWER COMMENTS

We would like to thank the reviewers for their comments as they have significantly helped to improve our manuscript. Below, we provide a point-by-point response to each comment/question (blue text). All changes in the main text and supplementary files appear in red text in the revised version of this manuscript.

Reviewer #1 (Remarks to the Author):

In this report, the authors used capped small RNA-seq (csRNA-seq), sRNA-seq, and total RNA-seq to identify and quantify genes/regions undergoing active transcription initiation in dry seeds and at different stages of germination. They also used ATAC-seq to analyze chromatin accessibility. In addition, the data were used to analyse TSS and other cis-elements. The study identified large sets of genes that were under active transcription, even in dry seeds.

The authors conducted careful and comprehensive analysis of the various transcriptomics data and ATAC-seq data. However, my main concern is the method used to identify newly initiated Pol II transcripts, which is flawed. This method compares csRNA-seq reads with sRNA-seq reads to identify enriched reads in the csRNA-seq data set as newly initiated Pol II transcripts. However, this method is prone to generating false positives, as the enriched reads may actually be cap-containing degradation products of steady-state RNAs. For instance, let us assume that all small RNAs in an RNA sample are degradation products of steady-state RNAs, with some of them being cap-containing fragments from the 5' region of RNAs. In sRNA-seq, these small RNAs would be all identified, and during csRNA-seq library construction, the cap-containing fragments would be enriched. By comparing sRNA-seq reads with csRNA-seq reads, the enriched reads would be identified as 'newly initiated Pol II transcripts,' although they are just degradation products of steady-state RNAs.

This method's flaw could be the reason why the authors identified a large number of genes undergoing active transcription initiation in dormant dry seeds, while the enriched cap-containing RNAs were possibly degradation products of stored mRNAs in the dry seeds. Dormant dry seeds are metabolically inactive organisms that do not provide the necessary microenvironment for many genes to be actively transcribed. If many genes undergo active transcription in dormant seeds, it would lead to significant loss of seed reserves, as cells would need to produce nucleotides, as well as proteins and enzymes to synthesize RNA building blocks and for the transcriptional process, severely damaging the seeds' function.

I understand that the csRNA-seq method was published in Genome Research by another group four years ago and that the authors of this manuscript have invested a lot of effort in conducting the experiments and analyzing the data. However, I would reconsider my view on the report if the authors can show that my concerns about the method are unfounded.

Thank you for your comments. We understand that the csRNA-seq method that we chose for our study is relatively new, and this may generate some concerns. In this revised version, we provide more information in order to clarify the issue of possible contamination by RNA degradation products. We have also re-analyzed data and conducted additional experiments to support our view that the use of the csRNA-seq method is well founded for detection of RNAPII initiated transcripts. Below, we provide a point by point response to the different issues raised.

“...my main concern is the method used to identify newly initiated Pol II transcripts, which is flawed.”

The csRNA-seq method was first published in 2019 (Duttke et al Genome Research 2019, doi: 10.1101/gr.253492.119). Since then, csRNA-seq has been applied in a number of studies published in peer-reviewed journals (Lim et al J Exp Med 2021, doi.org/10.1084/jem.20202733; Shamie et al NAR Genomics and Bioinformatics 2021 doi.org/10.1093/nargab/lqab061; Duttke et al Front Neuroscience 2022, doi.org/10.3389/fnins.2022.858427; Branche et al, Nat Comms 2022, doi.org/10.1038/s41467-022-33041-1; Li et al PLOS Comp Biol 2023, doi: 10.1371/journal.pcbi.1010991) and pre-prints (de Jong et al Biorxiv 2023, doi.org/10.1101/2023.04.13.536694; Sloutskin et al Biorxiv 2023, doi.org/10.1101/2023.06.11.544490; McDonald et al Biorxiv 2023, doi.org/10.1101/2023.09.25.559415). Given its sensitivity to detect unstable RNAs, we will probably see more use of this method in the future.

Recently, csRNA-seq has been evaluated for its suitability to identify enhancer RNAs (eRNAs) in comparison to twelve other genome-wide RNA-seq assays including GRO-cap, NET-CAGE, STRIPE-seq and PRO-seq (Yao et al Nat Biotechnol 2022, doi: 10.1038/s41587-022-01211-7). In the latter, csRNA-seq ranked second among the methods tested for its sensitivity to detect active enhancers in human cells. eRNAs are short-lived, non-coding RNA molecules which can serve as excellent marks of RNA Polymerase II (RNAPII) activity. In fact, an examination of the number of such potential events in our dataset (see SI Text section *“Highly unstable TSS initiation events occur at all stages of germination”*) revealed them to be present at all times during germination, including in the dry seed:

SI Text Figure 6: Stage-specific Unstable TSSs are detected in all time-points

Tabulation of the number of TSSs without any associated existing transcript annotation or detectable RNA-seq transcript by csRNA-seq cluster (Fig 2a).

In this previous plot, the number of “unstable” TSSs, or TSSs where there was insufficient RNA-seq signal across all time-points, including the exosome mutants, in each of our csRNA-seq clusters from Fig 2a. As these events are producing RNAs which are so unstable as to never be detected without the cap-enrichment step, it is likely that they only exist during the short period of initiation by RNAPII. Interestingly the sharp increase in such events in the C5 and C6 clusters (corresponding to the L26 and L57 time-points) suggests a total increase in transcription initiation upon the transition to post-germinative growth. Importantly however, these events are always being detected, including in the dry seed.

In addition, in one of the papers mentioned above (Shamie et al NAR Genomics and Bioinformatics 2021 doi.org/10.1093/nargab/lqab061) the authors performed a direct comparison of csRNA-seq and GRO-seq in the same set of samples clearly showing how comparative these two methods are to detect RNAPII nascent transcripts (Figure 1c; Supplementary Figure S2). With all this, we would like to highlight that recent work by experts in the field agree that the csRNA-seq method is an accurate methodology to detect RNAPII initiated transcripts.

“This method compares csRNA-seq reads with sRNA-seq reads to identify enriched reads in the csRNA-seq data set as newly initiated Pol II transcripts. However, this method is prone to generating false positives, as the enriched reads may actually be cap-containing degradation products of steady-state RNAs. For instance, let us assume that all small RNAs in an RNA sample are degradation products of steady-state RNAs, with some of them being cap-containing fragments from the 5' region of RNAs. In sRNA-seq, these small RNAs would be all identified, and during csRNA-seq library construction, the cap-containing fragments would be enriched. By comparing sRNA-seq reads with csRNA-seq reads, the enriched reads would be identified as 'newly initiated Pol II transcripts,' although they are just degradation products of steady-state RNAs. This method's flaw could be the reason why the authors identified a large number of genes undergoing active transcription initiation in dormant dry seeds, while the enriched cap-containing RNAs were possibly degradation products of stored mRNAs in the dry seeds.”

During the course of our analysis of the csRNA-seq data we did not detect any changes in characteristics for the dry seed-specific data which could suggest increased sequencing of RNA degradation products. We have included further analysis of the data to specifically address the point, which we have also included in SI Text. The added text is as follows:

“RNA degradation products are not a significant source of capped-small RNAs in dry seeds

It is generally understood that dry seeds do not undergo active transcriptional elongation as a consequence of their metabolically inert state. Despite this, previous studies have shown that they retain the potential for transcription via the presence of RNAPII in the nucleus (Comai & Harada, 1990; Zhao et al., 2022). As a result, it is logical to conclude that capped-small RNAs (which are the product of RNAPII transcription initiation) would be present within dry seeds, even if they are not being actively elongated. To test this, we examined the read distribution in TSSs and gene bodies in all csRNA-seq samples for evidence of increased RNA degradation products which could suggest a lack of RNAPII transcription initiation-specific products. We first calculated the ratio of reads within genes which were present specifically near the TSS to the entire gene and found that nearly all reads in all csRNA-seq libraries were present within the TSS region (SI Text Figure 4a), indicating successful enrichment of capped-small RNAs. Repeating the analysis with the input small RNA libraries showed that most detected small RNAs present within gene bodies were not originating from the TSS, though there was increased variability across time-points (SI Text Figure 4b). Crucially, the dry seed samples did not indicate increased ratios of reads present in the TSS relative to other samples, suggesting these samples did not have a specific increase in TSS-specific degradation products which could generate additional false-positive TSS peaks. We also compared these read counts to their total library sizes and observed largely similar patterns, with typical levels of relative read counts within the TSS regions in both capped-small and input RNA dry seed libraries (SI Text Figure 4c, d).”

SI Text Figure 4: Relative csRNA-seq read abundance in TSSs and gene bodies

(a), (b) Boxplots of the ratio of reads present in the TSS region of genes to the entire gene region in the capped-small and input RNA-seq libraries ($n = 19,688$). A value of 1 indicates that all reads over a gene are present within the TSS region. Some TSSs overlap regions outside of the gene and thus some ratios are greater than 1. The lower, middle and upper hinges correspond to first quartile, median, and third quartile, respectively. The lower and upper whiskers extend to the minimal/maximal value respectively or 1.5 times the interquartile range, whichever is closer to the median.

(c), (d) Barplots of the fraction of total reads in the capped-small and input RNA-seq libraries present within TSS and gene regions ($n = 19,688$). The lower, middle and upper hinges correspond to first quartile, median, and third quartile, respectively. The lower and upper whiskers extend to the minimal/maximal value respectively or 1.5 times the interquartile range, whichever is closer to the median.

We agree that capped degradation products of the correct size would be enriched in any of the csRNA-seq libraries. However, we would argue that these are generally not a significant contributor to TSS peak discovery and quantification, nor that the dry seed samples are more prone than others to false positive peaks. Not only was the RNA extraction method we used specifically optimized to allow for high RIN (>9) RNA samples for our dry seed samples (see Methods section), we did not observe that the csRNA-seq data contained large differences in quality (or other characteristics) between the dry seed and seedling samples.

Response letter Figure 1: The Bioanalyzer profiles of our two dry seed replicates [nDS1, nDS2] and our two seedling replicates [57H1, 57H2]. csRNA, sRNA, and RNA-seq libraries were prepared from the same tube of RNA for each sample/replicate.

In addition to our previous global analysis of the data (SI Text Figure 4), we would like to demonstrate a closer examination of the data. For example, we can clearly observe differential accumulation of TSS-originating capped RNAs specifically in the csRNA libraries, and conversely, non-TSS-originating uncapped RNAs specifically in the sRNA (input) libraries for both dry seed and seedling samples over *MIR390a*:

Response letter Figure 2: In this figure, TSS-specific enrichment is clearly observed in both dry seed and seedling csRNA-seq libraries only, whereas miRNA-specific enrichment is observed in both dry seed and seedling input libraries only. For both csRNA-seq and sRNA-seq only the 5-prime position of reads is maintained during the generation of the genome browser files.

Generally for most genes reads in input libraries were evenly distributed along the gene as seen in the following example:

Response letter Figure 3: csRNA-seq and sRNA-seq coverage over two less-expressed genes.

However, we could indeed observe an increase in TSS-like peaks in input libraries of very highly expressed genes (in addition to increased gene-body read coverage), such as *ACTIN7*:

Response Letter Figure 4: csRNA-seq and sRNA-seq coverage over *ACTIN7*.

Again, we stress that the characteristics of the dry seed data are not fundamentally different from the data of other samples. Thus, even if capped degraded RNAs are being falsely enriched in the csRNA-seq samples, this is an issue which occurs evenly across all samples. In fact we do not believe this type of issue is one unique to the csRNA-seq: indeed, any nascent transcriptomic method which involves an enrichment step (e.g. BrU-RNA immunoprecipitation for GRO-seq/cap, RNAPII immunoprecipitation for pNET-seq, etc) risks sequencing unintended RNAs. Additionally, we believe it could be argued that high levels of transcription initiation could generate sufficient levels of TSS-originating small RNAs to be detectable in the input libraries, meaning they may not actually be purely representing degradation products. If the *ACTIN7* mRNA was being degraded to such a severe extent, then it would be logical to expect a large increase in signal over the entire gene body, not so specifically at the TSS, however this is clearly not the case.

Furthermore, adjusting the scale of read densities across *ACTIN7* to reveal the placement of individual reads shows the same pattern of scattered low-level coverage of reads in all libraries in both dry seed and seedling samples across the gene bodies:

Response letter Figure 5: Lower abundance csRNA-seq and sRNA-seq coverage over *ACTIN7*.

In conclusion, while we do agree that capped RNA degradation products are likely contributing to csRNA-seq TSS peaks, they are likely only a minor contributor observable for only the most highly expressed genes. Critically, this occurs equally across all samples.

“Dormant dry seeds are metabolically inactive organisms that do not provide the necessary microenvironment for many genes to be actively transcribed. If many genes undergo active transcription in dormant seeds, it would lead to significant loss of seed reserves, as cells would need to produce nucleotides, as well as proteins and enzymes to synthesize RNA building blocks and for the transcriptional process, severely damaging the seeds' function.”

We thank the reviewer for their comment on this point. We agree that the seed microenvironment may not provide enough resources for active transcription elongation. Despite this, previous work has suggested that some level of active transcription may be taking place in dry seeds in Brassica species (Comai & Harada PNAS 1990; 10.1073/pnas.87.7.2671), the data provided in our work clearly demonstrate accumulation of RNAPII initiated transcripts in Arabidopsis dry seeds in the form of capped-small RNAs (csRNAs), which may not necessarily imply active RNAPII transcription.

We apologize if our interpretation of our results has led to misunderstanding. We believe that we have been able to detect active RNAPII transcription at early stages of imbibition, but we do not provide evidence for active transcription in dry seeds. In order to clarify this critical point we have done the following edits to the main text: abstract (lines 16-17), we have removed *“including in dry seeds”* from the previous statement *“Combining csRNA-seq, ATAC-seq and smFISH in Arabidopsis thaliana we demonstrate that active transcription initiation is detectable during the entire germination process including in dry seeds”*. Also within the main text (lines 77-78 changes in bold):

“By applying single molecule imaging, we demonstrated that active RNA synthesis takes place at all stages of germination, **from early imbibition onward.**”

As stated in our reply to a previous comment, we are confident that the csRNAs detected in our study are not degradation products. Thus, we have been able to capture capped-small RNAPII initiated transcripts in our dry seed samples. One possibility is that these csRNAs in dry seeds are the products of RNAPII transcription initiation and pausing during seed maturation, that have remained in the nucleus attached to RNAPII and to the gene locus during dehydration. Alternatively, we could also be detecting to some extent products of active RNAPII transcription initiation (although happening at very low rate). At this point, we are unable to discriminate between these two possibilities. To improve our manuscript, we have edited the Discussion section (lines 473-485) taking into account these two possible scenarios.

Noteworthy, either possibility (accumulation of paused RNAPII or active transcription initiation in dry seeds) would require accumulation of RNAPII at the 5' end of genes. To validate the presence of RNAPII over genes within seeds we performed RNAPII ChIP-qPCR targeting *ACT7* and *DOG1* in both dry and imbibed seeds. In both cases we observed significant enrichment of RNAPII near the TSS of each gene when compared to background levels in the genome (SI Text Figure 5a, b and below), demonstrating that RNAPII is present in the expected location within genes to generate initiated csRNAs of the appropriate size to be enriched in the csRNA-seq. The description of the ChIP-qPCR experiment has been added to SI Text (see section “RNAPII is present over gene bodies in seeds”).

SI Text Figure 5: RNAPII ChIP-qPCR in dry and imbibed seeds

(a) RNAPII ChIP-qPCR of dry (DS, n = 3) and imbibed (72 h stratified, S72, n = 4) seeds using primers targeting the *ACTIN7* gene (*Act7*; AT5G09810) obtained from Wu et al. (2016). Input-normalized RNAPII enrichment levels for each sample were normalized to

enrichment levels over the promoter of *Act7* (*Act7*_995). Statistical significant enrichment of RNAPII over background levels was determined by comparing the enrichment values with those obtained from primers targeting Intergenic Region 5 (IGN5, IGN5_Set1) obtained from Wu et al. (2016), and performing a one-sided Student's T-test ($P < 0.1$, $*P < 0.05$, $**P < 0.01$). Error bars indicate the SEM.

(b) RNAPII ChIP-qPCR of dry (DS, $n = 3$) and imbibed (72 h stratified, S72, $n = 4$) seeds using primers targeting the *DOG1* gene (AT5G45830) obtained from Chen et al. (2020). Statistical enrichment of RNAPII over background levels was determined as done for (a).

ChIP-qPCR experiment showed that accumulation of RNAPII matches csRNA enrichment, reinforcing that csRNAs are products of RNAPII initiation rather than RNA degradation products.

Interestingly, we observe accumulation of nuclear transcripts in dry seeds. In Figure 1d, smRNA-FISH experiments revealed nuclear accumulation of the *AT1G04170* transcript from early imbibition and up to the seedling stage, confirming that csRNA-seq was capturing true transcriptional activity. During this revision, we have performed additional smRNA-FISH experiments to evaluate accumulation of this transcript in dry seeds. Despite the high background noise and weak signal, using the same set of exonic and intronic probes (Figure S1e and below) as in Figure 1D, we were able to detect accumulation of full-length *AT1G04170* in nuclei in dry seeds. Although this evidence is still not enough to rule out whether there is active RNAPII transcription initiation/elongation or accumulation of paused capped transcripts in Arabidopsis dry seeds, we are confident that the csRNA-seq is more likely capturing chromatin/nuclear accumulated transcripts rather than RNA degradation products. Since we do not claim to provide evidence of active transcription in dry seeds in our manuscript and do not wish to confuse the issue, we have decided to only show the smRNA-FISH data in dry seeds in this response letter, and not to include it in our manuscript.

Fig S1e: Showing the csRNA-seq and RNA-seq of *AT1G04170* in dry and 72 h stratified seeds, as well as seedlings. Also shown are the locations of the probes used for smFISH.

Response letter Figure 6: smFISH in squashed dry seeds using probes targeting the unspliced mRNA of the gene *AT1G04170*, exactly as described for Fig 1d, Fig S1d, e. White arrows point to sites of transcription in the nucleus, which may not be necessarily active. Nuclei were stained with DAPI, pseudo-coloured in blue. Scale bar = 5 μ m.

Ruling out whether there is active RNAPII transcription initiation/elongation or accumulation of paused capped transcripts in Arabidopsis dry seeds is a very exciting aspect of seed biology that goes beyond the present study.

Reviewer #2 (Remarks to the Author):

This manuscript by Tremblay and colleagues describes an interesting and important study into nascent transcription during germination. The context of the study - that we currently understand little about how and when transcription is initiated during germination - is a significant unknown in the field of germination biology. To address this the authors apply capped small (cs) RNA-seq, a method that identifies RNAs at a very early stage of transcription (2-60 nt) and consequently allows transcriptional start sites to be mapped. In parallel the authors generate accessible chromatin maps and traditional RNA-seq. All of these data are generated across a time series of germination from dry seeds to young seedlings. Through a series of clever analyses the authors make a number of observations that substantially advance our understanding of seed germination. These include detecting active transcription in dry seeds - essentially prior to germination - and describing genome-wide transcription of many long-non-coding RNAs in germinating seeds. They illustrate how nascent transcription and chromatin accessibility data can be used to define gene regulatory programs active during germination and identify candidate transcription factors. This analysis then allows them to define more clearly the molecular mechanism for ABA sensitivity in seeds that have commenced germination, which is due to continued promoter accessibility in ABA response genes after their transcription has begun to decrease. The authors also investigate properties of antisense transcripts in germination, deducing that antisense and sense transcripts from individual genes are likely to exist in different cell types because the sense transcript appears to disrupt transcription of the antisense transcript. Bidirectional promoters are another area of focus, where the authors use their data to create an extensive resource of promoters driving different combinations coding and

non-coding loci and to understand where these are or are not co-ordinately expressed. The study of bidirectional promoters is notable in that there is controversy in the field about their existence; however, I did at times struggle to follow this section of the manuscript and have some suggested revisions to aid clarity.

This work substantially advances our knowledge of the processes regulating gene expression in germinating seeds, and has broad/general implications for gene regulation in Arabidopsis. Prior to this study, we did not know how early transcription was initiated during germination, nor how extensive non-coding transcription was. Overall I am very impressed and feel the manuscript merits publication in Nature Communications with some minor revisions, details of which follow. These revisions should only require work on the manuscript text and possibly some minor reanalyses of existing data.

Signed, Mat Lewsey.

We sincerely thank the reviewer for their positive comments and are encouraged by their enthusiasm for our work. We have addressed each suggestion/comment point by point.

Suggested revisions and points for clarification

- Fig. 1c - in legend clarify if genes are ordered the same in each panel.

We have added the following sentence to the Fig 1c legend:

“Rows (genes) were ordered in descending order by their total signal strength in the csRNA-seq.”

- Fig. 1d - what is the interpretation of the foci of unspliced transcript outside of the nucleus?

We believe these represent the gradual accumulation of the mature mRNA product due to continuous nascent transcription occurring during the entirety of the germination process. It is likely that due to a sufficient number of probes (28/48) over exonic regions, these are still visible in the smFISH (though with an obvious decrease in brightness). To clarify this point we have added the following text to the Fig S1d legend (additions in bold): “Brighter spots in the nucleus (see arrows) represent active transcription sites in germinating seeds, **whereas smaller bright dots in the surrounding area likely indicate spliced transcripts bound by a smaller number of exonic-only probes (28 / 48 total probes).**”

- Ln 196. 'however none of the TFs belonging to the C4 cluster had similar binding sites (Dataset S7). ' I got lost here. Do you mean expressed in rather than belonging to?

We were attempting to convey that the cluster 4 TSSs were enriched for certain binding sites, but we could not match these binding sites to known motifs associated with transcription factors whose TSSs were assigned to cluster 4. We have attempted to clarify the text (line 198) thusly (changes in bold):

“Two of the most highly enriched cis-elements (M1 and M4) were matching with ABI5 and RELATED TO AP2 1 (RAP2.1), both of which are expressed and active within the DS. Also highly enriched were the Telo-box (M23) and Site II motif (M2) in L6-associated clusters (mostly in translation-associated genes), however none of the TFs **expressed in** the C4 cluster had similar binding sites (Dataset S7).”

- Ln 226. 'as a nucleosome can be observed in the promoters of DS-specific genes and more generally at instances of the motif itself'. Expand on this to be more clear about what you think that nucleosome indicates/is doing.

We have attempted to clarify the text (lines 229-230, changes in bold):

“Using a previously published MNase-seq dataset from leaves confirmed this, as a nucleosome can be observed in the promoters of DS-specific genes and more generally at instances of the motif itself, **likely blocking access for TFs and RNAPII to initiate transcription** (Figure S3f-g)⁴¹.”

- Ln 296. 'the presence of a proximal antisense TSS generally led to lower levels of expression'. Correlated with, not led to - causality is not demonstrated here.

We have added modified the text accordingly (line 298-301, changes in bold):

“Furthermore the presence of a proximal antisense TSS generally **correlated with** lower levels of expression of the sense TSS as well as reduced levels of RNAPII accumulation over the gene body when compared to the high levels of RNAPII detected in genes with distal antisense TSSs (Figure 5d; Figure S6e-g).”

- Ln 312. 'these data suggest that sense and antisense transcription likely cannot occur simultaneously and only appear to correlate in our bulk sequencing data as a result of their expression in different tissues.' If this is the case, what do you propose is the function of the antisense transcripts? Previously I thought they were assumed to act in feedback regulation of the sense transcript within the same cell.

As we have suggested within the text, we believe the trends we observed in our data suggest the most common mechanisms by which antisense transcription regulates sense/protein-coding transcription is by replacing it (i.e., suppressing it). However we are aware that some lncRNAs have been reported to positively regulate sense transcription within the same cell, e.g. the lncRNA *MAS* enhances deposition of H3K4me3 within the body of *MAF4* to increase its expression (Zhao et al. Nat Commun., 2018; 10.1038/s41467-018-07500-7). As such, we would like to stress that this was only the general trend in our data, and not a universal consensus among all

detectable sense/antisense pairs. Indeed, for many such pairs our calculated Pearson correlation coefficients were closer to zero, meaning we cannot discount that they are in fact representing more complex modes of sense/antisense regulation, perhaps with interchanging transcriptional states within the same cells over time. We believe an in depth discussion of this point to be beyond the scope of this study, so we wish to add the following brief mention in the discussion (lines 513-516): “However it is important to note that this may simply be the most common observable form of sense-antisense transcriptional regulation, as examples of positive regulation of sense transcription by the antisense lncRNAs have been reported (Zhao et al., 2018).”

- Fig 6d. Explain in left most panel, cs-RNA-seq, what the red and blue vertical lines indicate.

In the Fig 6d csRNA-seq heatmap antisense signal is represented using red and sense signal using blue. Each row is centered on an individual TSS at position 0 thus giving the appearance of a vertical blue line, when in fact it is simply a column of aligned individual blue squares. The rows were ordered in ascending order by the distance to their upstream antisense divergent TSS, thus again giving the appearance of there being a red line even though it is simply all of the individual antisense signals aligning. (Please see the explanation in the next comment response for more detail.)

- Ln 347. I got quite confused in this section. 'While there was a higher proportion of correlating ncTSS-pcTSS pairs than not, these still represented a minority of all discovered cases.' I'm uncertain what you mean here - please rephrase. In the relevant figure, 6d, why has row normalization been applied non-uniformly within rows (eg. csRNA-seq, legend says normalization was applied from 1- to 1, when scale is -2 to 1, which will have differing effects on interpretation of every row in the plot since the TSSs are at different points relative to this scale)? What is the value of n in fig 6d? It appears from fig 6d that every nc/pc pair is expressed in an anti-correlated manner, but that's not what is described in the main text. Is that what you mean by divergent and somehow you have selected for these cases in your analysis? If not, please define divergent. And why is the hen2-4 data presented for RNA-seq - please explain the significance of that mutant here.

We apologize for the confusion we have engendered. We have attempted to clarify the text in the following manner (lines 349-351):

“While there was a higher proportion of correlating versus anti-correlating ncTSS-pcTSS pairs, correlating pairs still represented a minority of all discovered cases with most pairs having Pearson correlation coefficients less than 0.25.”

We apologize for the unclear description of Fig 6d. In this figure, we show the signal from various sequencing datasets in the region around protein coding TSSs with a detected divergent TSS (n = 1,643 as seen in Fig 6a; we have added this to the legend). Each row of each matrix is a single TSS region, with the TSS being a position 0 along the x-axis. From this point, the region is extended 1000 bp downstream (shortened to 1

in the axis labels) and extended 2000 bp upstream (shortened to -2 for 2000 bp, and -1 for 1000 bp in the axis labels). As csRNA-seq and RNA-seq provide strand-specific information, we normalized the sense strand read density within each TSS region (row) between 0 and +1, and the antisense strand read density between 0 and -1. Since the ATAC-seq, RNAPII ChIP-seq and MNase-seq do not provide strand-specific information, the total read density within each TSS region (row) is normalized between 0 and 1. These two types of scales are represented as the red/white/blue scale for strand-specific normalized data, and purple/yellow for non-strand-specific data. Perhaps the fact that the colour scales and the x-axis labels are the same values, despite the fact that they represent different things (one is signal value, the other is distance from the TSS) is leading to their being confused for each other; we apologize for this. The intent of this normalization is only to visually demonstrate the colocalization of sense and antisense csRNA-seq signal with corresponding RNA-seq, ATAC-seq, RNAPII ChIP-seq and MNase-seq signal, and not to show whether the protein coding and divergent TSSs are correlated or their level of expression. The inclusion of *hen2-4* instead of Col-0 for the RNA-seq heatmap was due to the fact that divergent transcripts are highly unstable and underrepresented in the Col-0 RNA-seq, whereas in the *hen2-4* RNA-seq their increased abundances provide a clearer picture of the concordance between transcription initiation (csRNA-seq) and transcription elongation (RNA-seq).

We have included a diagram to provide an alternative visual explanation:

We have added this figure to Fig S7 with the following legend entry:

“(j) Diagram explanation of the process of generating the heatmaps in Fig 6d. Divergent TSS regions across the genome are first collected in 3 Kbp chunks (2 Kbp upstream and 1 Kbp downstream of the pcTSS) and sorted in ascending order by the distance between the pcTSS and their divergent TSS (Step I). Then, the sense and antisense read densities are individually normalized between 0 to 1 and 0 to -1, respectively (Step II). Finally, all normalized signal vectors are assembled vertically and plotted in the style of a heatmap (Step III).”

Additionally, we have added the following sentence to the Fig 6d legend entry to guide reads to this supplementary figure:

“A diagram explaining the process of generating these heatmaps in more detail can be found in Fig S7j.”

- More broadly, I think the importance / biological relevance of the exosome mutants has not been explained in enough detail for readers from outside the non coding RNA discipline.

We thank the reviewer for this suggestion. We have added more text to the SI Text section “*The csRNA-seq captures transcription initiation independent of transcript stability*”:

“Due to the generally inherently unstable nature of non-coding RNAs, these are generally captured in lower abundances in typical RNA-seq experiments. Since our aim with the csRNA-seq was to faithfully capture the levels of genome-wide transcription initiation irrespective of transcript stability during our time-course, we wished to test whether we could observe an under-enrichment of such RNAs in our csRNA-seq datasets. To do this, we repeated the csRNA-seq and RNA-seq experiments using *hen2-4* and *rrp4-2* mutant plants, which accumulate higher levels of unstable non-coding RNAs due to defects in their RNA degradation pathways. Using the non-coding transcriptome data we obtained from these samples, we examined whether these could inform us as to the contribution of cytoplasmic RNAs (as opposed to nascently transcribed RNAs) to the csRNA-seq quantification. ... In conclusion, the lack of consensus between the RNA-seq and csRNA-seq expression levels of those non-coding RNAs we found to be unstable suggest there is no association between transcript stability and signal abundance in the csRNA-seq.”

However due to space constraint limitations we have not provided this explanation in the main text. We have instead added a small clarification to the main results (lines 99-100, in bold):

“In order to allow us to reconstruct additional putative lncRNAs associated with novel transcription start site (TSS) clusters from our csRNA-seq, we additionally sampled mutants in genes related to this pathway including *HUA ENHANCER2* (*hen2-4*) and

RIBOSOMAL RNA PROCESSING 4 (rrp4-2) at the fully expanded green cotyledon stage, **confirming the capture of unstable non-coding RNAs by our csRNA-seq (see SI Text)^{28,29}.**

- Fig 7a. As with 6d, what do the red and blue vertical lines indicate in the csRNA-seq panel? If I apply the scale bar from the right to them, it suggests to me that all these sites are anticorrelated. I don't understand the non-uniform row normalization within rows here nor the particular relevance of the *hen2-4* mutant here either.

We sincerely apologize for the confusion we have caused due to our unclear figure descriptions. Please see the explanation we provided for Fig 6d, as both that and Fig 7a are generated in the same manner. The only difference is that the "0" point along the x-axis represents the center point between the TSSs in Fig 7a as opposed to the position of the protein-coding TSSs in Fig 6d. We have added the following text to the Fig 7a legend entry:

"These heatmaps were generated in a similar fashion to those found in Fig 6d, except centering the distance 0 point around the midpoint between the two TSSs."

- Fig 7b-d. Label the rows directly in your genome browser screenshots. 7f, states top and bottom 500 candidate enhancers in the legend, but active/inactive in the figure itself; are these the same thing, and how were inactive enhancers defined? This point applies to the related main text also.

We thank the reviewer for their suggestion. We have added the labels to Fig 7b-d. Yes, we do refer to the top/bottom 500 candidates as active/inactive, respectively. We have clarified this point in the legend and in the main text.

Within the legend:

"Read density heatmaps of various sequencing datasets for the top 500 and bottom 500 (by total signal intensity) candidate enhancers in the L57 sample."

Within the main text (lines 440-444), additions in bold:

"Enhancers active in the seedling stage (L57) from our list (**defined as the top 500 by total csRNA-seq signal**) were enriched for RNAPII and the active histone mark H3K9ac, whereas inactive enhancers (**defined as the bottom 500 by total csRNA-seq signal**) generally were less accessible and had higher levels of the repressive histone mark H3K27me3 (Figure 7f)."

- Supplemental information - detection of exosome sensitive transcripts is described here, but the significance is not explained in enough detail. Please expand on what this tells you.

We thank the reviewer for this suggestion. As discussed in the previous comment response regarding the exosome mutants we have expanded the description of their significance.

- This may demonstrate the limits of my understanding of nascent transcription profiling, but how do you tell whether this is active transcription or paused/poised polII at genes that will become transcribed later? Put another way, considering all of the 'active' TSSs you detected in dry seeds, how many had a clear RNA-seq signal (ie. full length transcript) at that same time point? Fig S1 provides a nice single example of where csRNA-seq and RNA-seq match up in the dry seed, but was this how it appeared at all csRNA-seq sites? What I am getting at is could polII be recruited to those promoters during late seed development and initiate transcription, then pause and sit there until germination, as opposed to your results showing active transcription in the dry seed, or can you rule that out?

We thank the reviewer for bringing up this important point. The csRNA-seq data alone does not allow us to differentiate between paused and unpaused initiated transcripts. However, an analysis of genes which undergo transcriptional pausing from pNET-seq data by Zhu et al. (Nat Plants, 2018; 10.1038/s41477-018-0280-0) revealed that pausing is likely only ever occurring for a minority of active genes at a time in Arabidopsis. As a result we believe that only a minority of reads within our csRNA-seq data could be originating from paused RNAPII. Importantly however, these experiments were performed in seedlings, which are transcriptionally active and not inert (such as dry seeds are). As a result we do agree that it is reasonable to assume that the extent of pausing is significantly different between dry seeds and later, more active, stages. In all likelihood, we also believe there is at least a necessarily very significant decrease in the total level of active transcription in dry seeds compared to later stages (though potentially not a complete lack of it; see Comai & Harada (PNAS 1990; 10.1073/pnas.87.7.2671)). In addition, as the reviewer suggests, we agree that RNAPII is being captured “frozen” mid-transcription of genes within the dry seeds (perhaps either to be resumed or stopped and disassociated upon rehydration). Indeed, examining the TSSs present within the dry seed-specific cluster in Fig 2a (cluster C1, n = 4,607), 284 are “unstable” TSSs with no detectable RNA-seq signal in any of our time-points, including the exosome mutants. This suggests these TSSs are sites of RNAPII transcription initiation producing RNAs so unstable they never accumulate to detectable levels during seed maturation. This may not be evidence that they are still being actively initiated in the dry seed, but we believe that at the very least it signifies RNAPII is physically present over these loci at the time of RNA extraction of these samples. Interestingly, expanding this analysis (see “*Highly unstable TSS initiation events occur at all stages of germination*” in SI Text) to all clusters reveals a trend whereby the count of unstable TSSs increases dramatically in the C5 and C6 clusters (i.e., the L26 and L57 time-points):

SI Text Figure 6: Stage-specific Unstable TSSs are detected in all time-points

Tabulation of the number of TSSs without any associated existing transcript annotation or detectable RNA-seq transcript by csRNA-seq cluster (Fig 2a).

Perhaps this is indicative of a sharp increase in total transcription initiation upon the transition to post-germinative growth.

To further test the theory that RNAPII is physically present over these loci in dry seeds at the time of sampling, we performed RNAPII ChIP-qPCR of *ACTIN7* and *DOG1* in dry and stratified seeds to confirm the presence of RNAPII near the TSS of these genes (see SI Text section: “*RNA degradation products are not a significant source of capped-small RNAs in dry seeds*”). From this we concluded that even if the RNAPII is not actively elongating, it is present in sufficient quantities in the dry seed in an “initiated” fashion to have generated the capped-small RNAs we are observing in our csRNA-seq. (Please see the responses to reviewer 1 for additional comments on this topic.)

Addressing this question in further detail, as well as the topic of what happens generally to the RNAPII upon rehydration, is of great fascination to us, though outside the scope of this study.

SI Text Figure 5: RNAPII ChIP-qPCR in dry and imbibed seeds

(a) RNAPII ChIP-qPCR of dry (DS, n = 3) and imbibed (72 h stratified, S72, n = 4) seeds using primers targeting the *ACTIN7* gene (*Act7*; AT5G09810) obtained from Wu et al. (2016). Input-normalized RNAPII enrichment levels for each sample were normalized to enrichment levels over the promoter of *Act7* (*Act7_-995*). Statistical significant enrichment of RNAPII over background levels was determined by comparing the enrichment values with those obtained from primers targeting Intergenic Region 5 (*IGN5*, *IGN5_Set1*) obtained from Wu et al. (2016), and performing a one-sided Student's T-test ($P < 0.1$, * $P < 0.05$, ** $P < 0.01$). Error bars indicate the SEM.

(b) RNAPII ChIP-qPCR of dry (DS, n = 3) and imbibed (72 h stratified, S72, n = 4) seeds using primers targeting the *DOG1* gene (*DOG1*; AT5G45830) obtained from Chen et al. (2020). Statistical enrichment of RNAPII over background levels was determined as done for (a).

Reviewer #1 (Remarks to the Author):

1. In the previous version of the manuscript, my main concern was regarding the conclusion that many genes in dry seeds were under active transcription, which was one of their major findings. My concern was on using the csRNA-seq method to draw the conclusion in dry seeds. I did not have the same concern regarding their conclusions about the nascent transcriptomes in imbibed seeds or germinating seedlings, which were also analyzed using the same methods.

The csRNA-seq method identifies short cap-containing transcripts. It has been used in several studies since its initial publication for identifying transcription start sites (TSSs), transcription regulatory elements, enhancer RNAs, and promoter upstream transcripts. It can also reveal actively transcribing regions.

However, as I mentioned in my previous comment on the manuscript, the csRNA-seq reads observed in dry seeds may not necessarily be from actively transcribed genes. Instead, they could be derived from partially degraded transcripts and/or paused transcripts (as the other reviewer also noted). This issue is not a concern when profiling nascent transcriptomes by csRNA-seq in tissues that are metabolically active and degrading transcripts and paused transcripts only account for a small portion of the transcriptomes. However, it is generally believed that dry seeds, being metabolically inert, are unlikely to undergo active transcription of many genes. In dry seeds, partially degraded transcripts and paused transcripts could represent a significant portion of the transcriptomes for many genes. Consequently, if the csRNA-seq data were interpreted without considering this, these transcripts could be misidentified as "nascent transcripts." The authors repeatedly referred to the publication by Comai and Harda (PNAS, 1990) as evidence that some genes in dry seeds could be under active transcription. However, in the Comai et al. study, they demonstrated that nuclei isolated from dry seeds could transcribe some genes (i.e. they are "transcriptionally competent"). It is important to note that the conditions and microenvironments of the isolated nuclei are very different from those in dry seeds. Comai and Harada stated in their paper that "we interpret the results to indicate that the genes are transcriptionally competent, although they are probably not transcribed actively in vivo in a quiescent dry seed." Therefore, the authors need to cite this paper accurately.

In the revised manuscript, the authors have largely addressed my main concern although they still suggested that many genes in dry seeds could be under active transcription. They might need to mention the possibility that when tissues are not undergoing active transcription like in dry seeds, partially degraded and paused transcripts could be misidentified as nascent transcripts in this type of analysis.

2. The authors presented data on the identification of bidirectional non-coding promoters (ncTSS-ncTSS). In animals, many enhancer RNAs are transcribed unidirectionally. Do the authors also have data on unidirectional non-coding promoters?

3. It has been discovered that many previously identified "non-coding RNAs" actually code for small peptides. It would be beneficial for the authors to compare their non-coding reads with recent literature on the genome-wide identification of small peptide-coding genes that were previously considered non-coding.

4. I find the following sentence a bit confusing: "Additionally, we observed that genes in which there was a positive correlation between the transcription of sense and antisense TSSs had lower average expression (Figure 5d)." In my understanding, a "positive correlation" implies that these genes have (1) high levels of both sense and antisense transcripts or (2) low levels of both sense and antisense transcripts. However, if it is the case of (1), how do the authors define that these genes have "lower average expression"?

Reviewer #2 (Remarks to the Author):

This is a second round review. I appreciate the significant effort the authors have made to revise the manuscript. I am satisfied that all of my comments are fully addressed.

RESPONSES TO REVIEWER COMMENTS

We would like to again thank the reviewers for their comments as they have significantly helped to improve our manuscript. Below, we provide a point-by-point response to each comment/question (blue text). All changes in the main text and supplementary files appear in red text in the revised version of this manuscript.

Reviewer #1 (Remarks to the Author):

1. In the previous version of the manuscript, my main concern was regarding the conclusion that many genes in dry seeds were under active transcription, which was one of their major findings. My concern was on using the csRNA-seq method to draw the conclusion in dry seeds. I did not have the same concern regarding their conclusions about the nascent transcriptomes in imbibed seeds or germinating seedlings, which were also analyzed using the same methods.

The csRNA-seq method identifies short cap-containing transcripts. It has been used in several studies since its initial publication for identifying transcription start sites (TSSs), transcription regulatory elements, enhancer RNAs, and promoter upstream transcripts. It can also reveal actively transcribing regions.

However, as I mentioned in my previous comment on the manuscript, the csRNA-seq reads observed in dry seeds may not necessarily be from actively transcribed genes. Instead, they could be derived from partially degraded transcripts and/or paused transcripts (as the other reviewer also noted). This issue is not a concern when profiling nascent transcriptomes by csRNA-seq in tissues that are metabolically active and degrading transcripts and paused transcripts only account for a small portion of the transcriptomes. However, it is generally believed that dry seeds, being metabolically inert, are unlikely to undergo active transcription of many genes. In dry seeds, partially degraded transcripts and paused transcripts could represent a significant portion of the transcriptomes for many genes. Consequently, if the csRNA-seq data were interpreted without considering this, these transcripts could be misidentified as "nascent transcripts." The authors repeatedly referred to the publication by Comai and Harada (PNAS, 1990) as evidence that some genes in dry seeds could be under active transcription. However, in the Comai et al. study, they demonstrated that nuclei isolated from dry seeds could transcribe some genes (i.e. they are "transcriptionally competent"). It is important to note that the conditions and microenvironments of the isolated nuclei are very different from those in dry seeds. Comai and Harada stated in their paper that "we interpret the results to indicate that the genes are transcriptionally competent, although they are probably not transcribed actively in vivo in a quiescent dry seed." Therefore, the authors need to cite this paper accurately.

In the revised manuscript, the authors have largely addressed my main concern although they still suggested that many genes in dry seeds could be under active transcription. They might need to mention the possibility that when tissues are not undergoing active transcription like in dry seeds, partially degraded and paused transcripts could be misidentified as nascent transcripts in this type of analysis.

We thank the reviewer for their comments on the manuscript. We have endeavoured to clarify and amend our discussion of the dry seed csRNA-seq, though we apologize for our misuse of the Comai & Harada citation. We have modified all text which makes reference to Comai & Harada to emphasize that their work revealed dry seeds maintain “transcriptional competence”, and is not evidence of active transcription. We have also added text to both the results and discussion to emphasize that the dry seed csRNA-seq may be detecting previously degraded or initiated transcripts from seed maturation, which we hope sufficiently address the remaining concern of the reviewer. We do however wish to continue to mention that we cannot disprove that some, perhaps very rare, level of active transcription could be happening in dry seeds, even if that is not the main contribution to detected TSSs in the csRNA-seq.

Text modifications in the results (changes in bold): “Strikingly, we observed clear transcription sites within the nuclei of cells in all stages of germination (Figure 1d), confirming that the csRNA-seq is capturing true transcriptional activity and that transcription occurs even during the earliest stages of imbibition. **While we could also detect sites of transcription initiation in the DS csRNA-seq data these likely represent partially degraded or initiated RNAPII transcripts from seed maturation, though our data cannot disprove that some rare events of active transcription may be occurring at some TSSs (see SI Text).**” (lines 114-117)

Text modifications in SI Text (changes in bold): “It is generally understood that dry seeds do not undergo active transcriptional elongation as a consequence of their metabolically inert state. Despite this, previous studies have shown that they retain **some level of transcriptional competence** via the presence of RNAPII in the nucleus (Comai & Harada, 1990; Zhao et al., 2022). As a result, it is logical to conclude that capped-small RNAs (which are the product of RNAPII transcription initiation) would be present within dry seeds, even if they are not being actively elongated. ... In both cases we observed significant enrichment of RNAPII near the TSS of each gene when compared to background levels in the genome (SI Text Figure 5a, b), demonstrating that RNAPII is present in the expected location within genes to **have generated** initiated capped RNAs of the appropriate size to be enriched in the csRNA-seq.”

Text modifications in the discussion (changes in bold): “An interesting finding of our work is the detection of csRNA peaks widely distributed in dry seeds. Although a previous report suggested that there may be some level of **remaining transcriptional competence**¹¹⁵, metabolically inactive dry seeds may also not provide the necessary microenvironment for many genes to be actively transcribed. Yet, the csRNA-seq data contain clear signatures of transcription initiation, with corresponding significant levels of RNAPII accumulating at such sites in both dry seeds and during imbibition (see SI Text). **This observation implies different scenarios, the most likely of them being that csRNA-seq may be detecting RNAPII transcripts initiated from seed maturation and retained at the site of transcription in dormant dry seeds. Alternatively, partially degraded RNAPII transcripts or even some extent of transcriptional elongation**

taking place in dry seeds, although probably at a very low rate, may be contributing to the csRNA-seq peaks detected in our study.” (lines 476-487)

2. The authors presented data on the identification of bidirectional non-coding promoters (ncTSS-ncTSS). In animals, many enhancer RNAs are transcribed unidirectionally. Do the authors also have data on unidirectional non-coding promoters?

We thank the reviewer for bringing up this very important point. In fact, we also believe the unidirectional non-coding TSSs may be evidence of unidirectional enhancers. During the process of compiling the list of candidate enhancer regions, we in fact included such TSSs. This is mentioned in the following text from the results, including citations to relevant work (with a new clarification in bold): “some enhancers may be only transcriptionally active on a single strand (**unidirectional**)^{53,54}.” (lines 434-435) Since a significant portion of the manuscript is already dedicated to the analysis of such TSSs (Figure 4, Figure S4), we believe we have already substantially addressed this point. This is further discussed in the discussion as well: “As enhancers can sometimes only show evidence of unidirectional transcription^{53,54}, expanding our list to intergenic regions with such non-coding csRNA-seq signal led us to assemble a final list of 1,891 putative enhancers active during germination, which were strongly enriched nearby transcription factors with correlating expression patterns (Figure S10f-g, j-k).” (lines 555-560) We hope the reviewer agrees we have sufficiently addressed this point and apologize if our use of unclear language in the results may have engendered this confusion.

3. It has been discovered that many previously identified "non-coding RNAs" actually code for small peptides. It would be beneficial for the authors to compare their non-coding reads with recent literature on the genome-wide identification of small peptide-coding genes that were previously considered non-coding.

We thank the reviewer for bringing up this interesting point. We agree this is a fascinating area of study for further understanding of lncRNA biology and are looking to expand our work into this area. While we did analyze the coding potential of our putative lncRNA sequences (Figure S4j), we did not specifically look for small peptides. We do however believe this type of analysis is outside the scope of the current work and would be better suited for a followup study, as our work focuses on the potential for non-coding transcription to be involved in nuclear regulation of the genome, and not at the level of post-translational regulation. We have added the following text and new citation to the discussion to highlight the potential of this exciting area of research: “**While our analysis did not uncover any coding potential among the putative lncRNAs (Figure S4j), we did not specifically test whether these sequences encode small peptides which may be of regulatory importance outside the nucleus¹¹⁸. Investigating this emerging regulatory aspect of non-coding transcription will be an interesting followup to this work.**” (lines 496-500)

4. I find the following sentence a bit confusing: "Additionally, we observed that genes in which there was a positive correlation between the transcription of sense and antisense

TSSs had lower average expression (Figure 5d)." In my understanding, a "positive correlation" implies that these genes have (1) high levels of both sense and antisense transcripts or (2) low levels of both sense and antisense transcripts. However, if it is the case of (1), how do the authors define that these genes have "lower average expression"?

We thank the reviewer for their comment and apologize for the confusion on this point. Our use of the word "correlation" is in reference to the overall pattern of expression across all samples, which could be both low and high at different times. Our intention was to convey that when the sense TSSs reaches a peak of expression (between pairs of sense-antisense TSSs with similar expression patterns), its maximum expression is on average lower than genes which don't have an antisense TSS with a similar pattern of expression. We believe this may be indicative of sense and antisense transcription being mutually exclusive (and thus not all cells in a sample can be actively initiating the sense TSS, lowering the maximum possible captured expression of a gene compared to genes which could have potential to be initiating in a higher number of cells). We have revised the sentence to emphasize this (changes in bold): "Additionally, we observed that genes in which there was a positive correlation between the **expression patterns** of sense and antisense TSSs had lower average **maximum** expression (Figure 5d)." (lines 308-310)

Reviewer #2 (Remarks to the Author):

This is a second round review. I appreciate the significant effort the authors have made to revise the manuscript. I am satisfied that all of my comments are fully addressed.

We thank the reviewer for taking the time to help us in improving our work and considering our revised manuscript.

Reviewer #1 (Remarks to the Author):

The authors have sufficiently addressed my main concerns.